# Lightcone Hamiltonian for Ising field theory I: $T < T_c$

Andrew Liam Fitzpatrick[1], Emanuel Katz[1] and Yuan Xin[2,3]

**1** Department of Physics, Boston University, Boston, MA 02215, USA
**2** Department of Physics, Yale University, New Haven, CT 06520, USA
**3** Department of Physics, Carnegie Mellon University, Pittsburgh, PA 15213, USA

## Abstract

We study 2d Ising Field Theory (IFT) in the low-temperature phase in lightcone quantization, and show that integrating out zero modes generates a very compact form for the effective lightcone interaction that depends on the finite volume vacuum expectation value of the $\sigma$ operator. This form is most naturally understood in a conformal basis for the lightcone Hilbert space. We further verify that this simple form reproduces to high accuracy results for the spectra, the $c$-function, and the form-factors from integrability methods for the magnetic deformation of IFT. For generic non-integrable values of parameters we also compute the above observables and compare our numeric results to those of equal-time truncation. In particular, we report on new measurements of various bound-state form-factors as well as the stress-tensor spectral density. We find that the stress tensor spectral density provides additional evidence that certain resonances of IFT are surprisingly narrow, even at generic strong coupling. Explicit example code for constructing the effective Hamiltonian is included in an appendix.

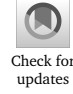

# 1  Introduction and summary

Lightcone (LC) Quantization is properly understood as an effective theory. It describes the degrees of freedom whose lightfront energies $P_+ = \frac{1}{\sqrt{2}}(E - p_x)$ become small in the limit of large momentum $p_x$, scaling like $\sim 1/p_x$. Indeed, the effective Hamiltonian describing these modes should itself exhibit this scaling, and consequently be invariant under boosts. In other words, the effective LC Hamiltonian is kinematically boost covariant, that is covariant with respect to the unperturbed boost transformation whose action on the effective theory states is to simply change their momentum label appropriately. This boost transformation leaves the trivial vacuum state invariant, and hence the trivial vacuum remains an eigenstate of the effective Hamiltonian. One may ask, how can this be, given the phenomenon of spontaneous symmetry breaking? In the perturbative limit, for theories with a Lagrangian description, this is intuitively possible because the proper effective Hamiltonian can have terms which explicitly depend on non-perturbative order parameters. In fact, when we use Feynman diagrams, we first determine all such terms (typically using the effective potential and vacuum equations of motion) before beginning perturbative computations. However, when we do not have a Lagrangian description and are at strong coupling, it is unclear if such an effective LC Hamiltonian exists in general. It is therefore useful to study a particular model which exhibits these kinds of complexities, where some quantities are well known from prior numerical work and integrability.

The 2d Ising Field Theory (IFT) is exactly such a model. It is the theory of a single 2d Majorana fermion with a thermal $\epsilon$ (aka 'mass') and spin $\sigma$ deformation:

$$H = \frac{1}{2\pi} \int_0^{2\pi R} dx \left( \psi \overline{\partial} \psi + \overline{\psi} \partial \overline{\psi} + m \varepsilon(x) + g \sigma(x) \right). \tag{1}$$

This theory is a non-Lagrangian theory, which in the low temperature phase (i.e. for $m < 0$) exhibits spontaneous $Z_2$ symmetry breaking as $g \to 0$. For moderate $g$ this theory has non-perturbative dynamics, leading to confinement of fermions, and several stable as well as meta-stable bound-states with physics resembling that of QCD. At very large $g$, the theory approaches a very interesting integrable $E_8$ limit which has been studied extensively, where all bound states become stable and there is no particle production [1]. It is hence an ideal laboratory for attempting to find the LC effective Hamiltonian, $(P_+)_{\text{eff}}$, and for studying its connection to the vacuum structure of the theory. In addition, in our prior paper [2], we have found evidence that $(P_+)_{\text{eff}}$ should exist for this model. In that work, we presented a non-perturbative formula for $(P_+)_{\text{eff}}$, which can be evaluated numerically for any value of $g$ and $m$ at any volume $R$ and checked that it captures the physics, agreeing with prior truncation and

integrability results. We also found numerically that the effective Hamiltonian indeed depends on the vacuum expectation value (vev) of $\sigma$ at finite volume.

In this work, our first goal is to develop an analytic expression for the effective Hamiltonian in the low temperature phase. As we will see, this expression is most easily derived using the Dyson series representation of the effective Hamiltonian:

$$(P_+)_{\text{eff}} = \lim_{t \to \infty(1-i\epsilon)} \left[ \frac{i\partial_t U(t)}{\langle U \rangle} \right]_{p_x \to \infty} = \lim_{t \to \infty(1-i\epsilon)} \left[ \frac{V(t)U(t)}{\langle U \rangle} \right]_{p_x \to \infty}. \tag{2}$$

We emphasize that $t$ here is lightcone time (i.e., $x^+$) since we want to obtain the lightcone $P_+$ Hamiltonian. $V(t)$ consists of the set of relevant deformations evolved with the CFT Hamiltonian, $U(t) = T\{\exp(-i\int_0^t dt' V(t'))\}$, while $\langle U \rangle$ is computed in the CFT vacuum. The precise prescription, which we spell out in more detail in section 2, requires that the large momentum limit $pR \gg 1$ be taken first. As we argued in our previous paper [2],[1] this effective Hamiltonian is to be evaluated in the chiral sector of the theory which constitutes the low energy degrees of freedom at large momentum. Moreover, it is most naturally computed in a conformal basis of states consisting of quasi primary operators of the CFT:

$$|l, p\rangle \equiv \int dx^- e^{-ip_- x^-} \mathcal{O}_l(x^-)|0\rangle. \tag{3}$$

The integral limits $-\infty$ and $\infty$ are omitted; integrals over lightcone spatial coordinates $x^-$ etc should be understood to be integrated from $-\infty$ to $+\infty$ unless otherwise specified. In this basis, the above effective Hamiltonian can be expressed in perturbation theory in terms of CFT connected correlators. For the specific case of very relevant operators, with dimensions $\Delta < 1$, as we will show, the Dyson series can be resummed to yield a non-perturbative expression which depends on the vevs of relevant operators. In particular, with only a $\sigma$ deformation this sum is given by

$$(P_+)_{\text{eff}} = g\langle\sigma\rangle \int dx^- (\sigma(x)\sigma(\infty) - \mathbb{1}), \tag{4}$$

where $\sigma(\infty) \equiv \lim_{x \to \infty} (x^2)^{\Delta_\sigma} \sigma(x)$ is the $\sigma$ operator conformally mapped to the point at infinity, and $\sigma(x)$ is implicitly at $x^+ = 0$. The vev of $\sigma$ depends on the volume, and can be found, for example, from the vacuum energy as $\langle\sigma\rangle = -\frac{\partial E_0(g,R)/R}{\partial g}$.

In the presence of both a $\sigma$ and an $\epsilon$ deformation, in the low temperature phase, the fermions are always confined, and it thus becomes plausible that including only the bosonic chiral sector should be sufficient to capture the dynamics. Hence *the $\sigma$ vev fully characterizes this phase*, and the effective Hamiltonian is given by

$$(P_+)_{\text{eff}} = \frac{m^2}{2} \int dx^- \psi \frac{1}{i\partial_-} \psi(x^-) + g\langle\sigma\rangle \int dx^- (\sigma(x)\sigma(\infty) - \mathbb{1}), \tag{5}$$

where $\langle\sigma\rangle = -\frac{\partial E_0(g,m,R)/R}{\partial g}$. We note that when $g$ becomes small, this Hamiltonian indeed captures the effects of spontaneous symmetry breaking by depending explicitly on the order

---

[1]The effective Hamiltonian constructed in [2] is similar to our previous definition in [3], but the earlier work was missing the crucial insight that the lightcone effective Hamiltonian must be obtained as the effective theory of the light sector of states that emerges kinematically from equal-time quantization in the limit of large momentum; in EFT terms, the cutoff scale of the heavy states being integrated out relative to the light states in the EFT is of order the equal-time momentum which is taken to infinity. We also showed in [2] that the improved definition matches a more algebraic approach based on taking the Schur complement of the equal-time Hamiltonian divided up into the appropriate heavy and light sectors; we show this equivalence explicitly up to fourth order in the coupling in appendix C.1.

parameter. We use the fact the $\epsilon$ deformation is the fermion mass term whose effective Hamiltonian can be obtained by integrating out the non-dynamical anti-chiral component. See [4] for a detailed derivation.

The fact that $g$ and $\langle\sigma\rangle$ appear together in the effective Hamiltonian only in the combination $g\langle\sigma\rangle$ means that, in practice, $g\langle\sigma\rangle$ ultimately just *sets the dimensionful scale of the theory* and therefore can be treated as an independent parameter that does not need to be known a priori. In fact, the only dimensionless input parameter is the combination $g\langle\sigma\rangle/m^2$, so that the lightcone Hamiltonian has the same number of input model parameters as the equal-time formulation in infinite volume.

Having determined the Hamiltonian, we proceed to numerically study the dynamics of this model using Lightcone Conformal Truncation (LCT). That is, we truncate the above conformal basis of states by considering only states up to some maximum dimension, $\Delta_{max}$, and numerically diagonalize the Hamiltonian in this sector.

We first compare our numerical results to those of $E_8$ integrability in section 3. We find excellent agreement for the spectrum, the precise dependence of the $c$-function, $c(\mu)$, on energy, and also for the momentum dependence of various bound-state form-factors. This is despite the fact that we are at a truncation which uses only several hundred basis states. Next, in section 4 we compute similar observables for generic model parameters in the low temperature phase. We compare these to boosted TCSA for various truncation and volume parameters. We find that even with tens of thousands of states, TCSA results depend on the volume and on truncation choices, adding additional uncertainty. For example, each of the two integrable limits (either near very large $|m|$ or near very large $g$) prefer different choices for the volume and boost momentum. LCT, however, is able to capture the entire parameter space of this phase, interpolating effectively between the integrable points. We also report on two novel measurements in the IFT model. The first, is the profile of the stress tensor form-factor in various stable bound-states. The second, is the spectral density of the stress-tensor above the two-particle continuum threshold. The spectral density displays the bound-state resonances of the model, which have a surprisingly narrow width, consistent with the prior findings of [5] and [6].

By contrast, the effective Hamiltonian (5) does not access the high temperature phase $T > T_c$. It is not clear from the present work what is missing, but it is likely that additional operators need to be included in the effective Hamiltonian to accommodate additional states corresponding to the fermions in the unbroken phase. A candidate of such operator is the disorder operator $\mu$, which obtains a vev in the high temperature phase. The new operator mixes the fermion and boson chiral sectors through OPE $\mu\psi \sim \sigma$. We leave this as an open question for future work.

## 2 Derivation of effective lightcone Hamiltonian

### 2.1 Effective Hamiltonian from Dyson series

In this section, we will derive the simple form (4) for the effective Hamiltonian due to the $\sigma$ deformation of the Ising CFT. For the sake of simplicity and concreteness, we will focus on the case with only a single relevant operator. Moreover, with a single relevant deformation, the argument will only make use of the fact that $\sigma$ has dimension $\Delta < 1$, so to emphasize this fact we will treat the general case of a 2d CFT deformed by a single relevant scalar primary operator $\mathcal{O}_R$ with dimension $\Delta$:

$$S = S_{\text{CFT}} + \frac{g}{2\pi}\int d^2z\, \mathcal{O}_R(z,\bar{z}), \quad V_R(t) \equiv \int_0^{2\pi} d\theta\, \mathcal{O}_R(e^{t+i\theta}, e^{t-i\theta}). \tag{6}$$

We take units where $R = 1$. In [2], we argued that the lightcone effective Hamiltonian can be obtained from the unitary evolution operator $U(t)$ from the limit in (2).[2] Because the argument from [2] is somewhat formal, in appendix C.1 we explicitly show up to $O(g^4)$ in perturbation theory how $(P_+)_{\text{eff}}$ from the Dyson series (2) reproduces an independent algebraic construction of $(P_+)_{\text{eff}}$, also in (2), where heavy states are integrated out by using the Schur complement of the heavy state sector in the full $P_+$ Hamiltonian.[3]

Now, starting from the expression (2), we obtain the compact formula (4) as follows. First, expand out the Dyson series for $U(t)$ evaluated between basis states (3) created with Fourier transforms of quasi-primary operators from the UV CFT:

$$\langle l|(P_+)_{\text{eff}}|l'\rangle = \sum_{n=0}^{\infty}\left(-i\frac{g}{2\pi}\right)^n \int_0^{2\pi} dx\,dz\, e^{ip(x-z)}\langle \mathcal{O}_l(x)V_R(t)\int_0^t dt_2 V_R(t_2)\cdots\int_0^{t_{n-1}}dt_n V_R(t_n)\mathcal{O}_{l'}(z)\rangle_{\text{conn}}, \tag{7}$$

where $\mathcal{O}_l(x)$ (and $\mathcal{O}_{l'}(z)$) should be understood to be inserted at any time greater than $t$ (less than 0, respectively). so that all the operators are time-ordered in the ordering as they appear. Only the connected part of the correlator contributes to the expression above. The reason for this is that we shift the Hamiltonian so that the vacuum energy of the interacting theory is zero:

$$V \equiv V_R - E_{\text{vac}}(g). \tag{8}$$

Then, the contribution from the disconnected piece (defined as the contribution where the bra and ket state just contract with each other, and not with any of the interaction terms) of the correlator vanishes:

$$\frac{\langle l|VU|l'\rangle_{\text{disc}}}{\langle U\rangle} = \frac{\langle l|(V_R - E_{\text{vac}})U|l'\rangle_{\text{disc}}}{\langle U\rangle} = \delta_{ll'}\frac{\langle(V_R - E_{\text{vac}})U\rangle}{\langle U\rangle} = \delta_{ll'}(E_{\text{vac}} - E_{\text{vac}}) = 0. \tag{9}$$

We treat this more thoroughly in appendix C.2 where we show in detail how the disconnected pieces cancel up to $O(g^4)$ in the Dyson series expansion.

Next, because $\mathcal{O}_l$ and $\mathcal{O}_{l'}$ are chiral operators, we can evaluate the correlator by summing over the singular terms in their OPE with the interactions $V_R$. At this point, we have to assume that the leading contributions to $(P_+)_{\text{eff}}$ at large $p_x$ all come from singular terms in the OPE when $\mathcal{O}_l$ and $\mathcal{O}_{l'}$ *both* approach the same factor $V_R$.[4] That is, we can perform an OPE of the

---

[2]The basic idea of the argument is that in the limit of large $p_x$, there is a separation of scales between states made entirely of particles with $p_x < 0$, whose $P_+$ energy scales like $p_x^{-1}$ at large $p_x$, and all other states, whose $P_+$ energy is $O(p_x^0)$ or greater. The effective lightcone Hamiltonian keeps the former ('light') states and integrates out the latter ('heavy') states. In order to integrate out the heavy states, one takes the lightcone time $t$ to be parametrically larger than the inverse energy of the lightest heavy states but smaller than the inverse energy of the light states, so that the phase oscillations of the heavy states are rapid whereas those of the light states are negligible. In [2], we removed states with rapid phase oscillations by averaging over a window in time, but a more convenient approach is to take $t$ large with a small imaginary piece, $t \to \infty(1-i\epsilon)$, so that rapid phase oscillations literally damp the contributions of the heavy states. Note that $t$ in (6) is the standard time coordinate, which seems to contradict the statement that in order for our procedure to extract the lightcone Hamiltonian $(P_+)_{\text{eff}}$ from $U(t)$, we should take $t$ to be the lightcone time which would instead enter as $V_R(t) = \int d\theta \mathcal{O}_R(e^{i\theta}, e^{-i\theta+2t})$ in order for the $t$ dependence to pick up the lightcone energy $\Delta - \ell = 2\bar{h}$ on the cylinder. However, the two expressions are equivalent when we take their matrix elements between any states, because the $d\theta$ integral forces the spin $h - \bar{h}$ of the bra state and the ket state to be the same, which implies that the difference in energy $h + \bar{h}$ of the bra state and ket state is the same as the difference in lightcone energy $2\bar{h}$.

[3]This general method was introduced into QFT Hamiltonian truncation studies in [7]. For prior uses in more general contexts, see the historical comment in [8].

[4]In fact, in the $\Delta > 1$ regime there is a more dominant term with $p_x^{\Delta-1}$ dependence and the argument breaks down. In the marginal case $\Delta = 1$, this term gives an IR divergence and lifts some states out of the spectrum. In this work, we use the fact that the $\epsilon$ deformation is the fermion mass term whose effective Hamiltonian is known and the IR divergence has been treated thorough, for instance see [4]. See Appendix D for a treatment of the IR divergence tailored to Ising Field Theory. A more general discussion of the $\Delta \geq 1$ singularity can be found in [9].

triple product:

$$\mathcal{O}_l(x)\mathcal{O}_R(y,\overline{y})\mathcal{O}_{l'}(z) \overset{x\sim y}{\underset{z\sim y}{=}} \frac{1}{(x-y)^{h_l}(z-y)^{h_{l'}}}f\left(\frac{z-y}{x-y}\right)\mathcal{O}_R(y,\overline{y})+\dots, \tag{10}$$

where $\dots$ are subleading in the limit $x\sim y, z\sim y$. The function $f$ of the ratio $\frac{z-y}{x-y}$ arises above because both $x$ and $z$ approach $y$ and so the argument of $f$ is not fixed by this limit. It will be more convenient to parameterize the leading term in the triple product OPE as

$$\mathcal{O}_l(x)\mathcal{O}_R(y,\overline{y})\mathcal{O}_{l'}(z) \overset{x\sim y}{\underset{z\sim y}{=}} \frac{1}{(x-z)^{h_l+h_{l'}}}m_{ll'}\left(\frac{z-y}{x-y}\right)\mathcal{O}_R(y,\overline{y})+\dots, \tag{11}$$

for some function $m_{ll'}$. By inspection, we see that we can extract this function from the following four-point function:

$$m_{ll'}\left(\frac{z-y}{x-y}\right)=(x-z)^{h_l+h_{l'}}\langle\mathcal{O}_R(\infty)\mathcal{O}_l(x)\mathcal{O}_R(y,\overline{y})\mathcal{O}_{l'}(z)\rangle_{\text{conn}}. \tag{12}$$

Applying this OPE to each of the $n$ $V_R$ factors in the Dyson series in (7), we obtain

$$
\begin{aligned}
\langle l|(P_+)_{\text{eff}}|l'\rangle =& \left(\int dx\,dz\, e^{ip(x-z)}\langle\mathcal{O}_R(\infty)\mathcal{O}_l(x)\mathcal{O}_R(y,y^{-1})\mathcal{O}_{l'}(z)\rangle_{\text{conn}}\right)\\
&\times g\frac{d}{dg}\sum_{n=0}^{\infty}\left(-i\frac{g}{2\pi}\right)^n\langle(V_R(t)\int_0^t dt_2 V_R(t_2)\cdots\int_0^{t_{n-1}}dt_n V_R(t_n))\rangle_{\text{conn}}.
\end{aligned}
\tag{13}
$$

By translation invariance, the first line is actually independent of $y$ and we can set $y$ to any value we want without loss of generality. Moreover, the matrix elements of the integral $\int dy\,\mathcal{O}_R(y,y^{-1})\mathcal{O}_R(\infty)$ between the definite momentum states $|l\rangle$ and $|l'\rangle$ are just the matrix elements of $\mathcal{O}_R(y,y^{-1})\mathcal{O}_R(\infty)$ at fixed $y$ times a momentum conserving $\delta$ function. Finally, the second line is simply $-g\frac{d}{dg}E_{\text{vac}}(g)=g\langle\mathcal{O}_R\rangle$. So we at last arrive at the compact result

$$(P_+)_{\text{eff}}=g\langle\mathcal{O}_R\rangle\int dy(\mathcal{O}_R(y,y^{-1})\mathcal{O}_R(\infty)-\mathbb{1}), \tag{14}$$

where $-\mathbb{1}$ removes the disconnected piece.

## 2.2 Effective Hamiltonian at second order

Although the above argument is nonperturbative, it will be illuminating to describe some of the steps in more detail when working at second order in the coupling $g$. Moreover, because all of the nonperturbative dependence on the coupling enters through the prefactor $\langle\mathcal{O}_R\rangle$, a second order analysis will actually tell us the entire effective Hamiltonian up to an overall constant. So in practice, this will also lead us to an algorithm for computing the effective Hamiltonian matrix elements starting from the functions $m_{ll'}$.

Begin with the following correlator on the plane with operators $\mathcal{O}_l$ and $\mathcal{O}_{l'}$ to create the basis states, and two insertions of the relevant operator $\mathcal{O}_R$:

$$\langle\mathcal{O}_l(x)\mathcal{O}_R(y_1,\overline{y}_1)\mathcal{O}_R(y_2,\overline{y}_2)\mathcal{O}_{l'}(z)\rangle. \tag{15}$$

We can conformally map this correlator to the cylinder, $w=e^{t+i\theta}, \overline{w}=e^{t-i\theta}$, in which case it picks up conformal factors $x^{h_l}z^{h_{l'}}(y_1\overline{y}_1 y_2\overline{y}_2)^h$. To pick out the momentum of the external

states $|l\rangle$ and $|l'\rangle$, we Fourier transform with respect to the angles of $x$ and $z$, respectively:

$$
\begin{aligned}
|l'\rangle &\equiv \frac{1}{N_{l,p}} \int d\theta \, e^{-ip\theta} \mathcal{O}_{l'}(e^{i\theta})|0\rangle = \frac{1}{N_{l,p}} \oint \frac{dz}{iz} z^{-p} \mathcal{O}_{l'}(z)|0\rangle, \\
\langle l| &\equiv \frac{1}{N_{l',p}} \int d\theta \, e^{ip\theta} \langle 0|\mathcal{O}_l(e^{i\theta}) = \frac{1}{N_{l',p}} \oint \frac{dx}{x} ix^p \langle 0|\mathcal{O}_l(x).
\end{aligned}
\tag{16}
$$

The factors $N_{l,p}$ are normalization factors, required since

$$
\langle l|l\rangle = \frac{2\pi \Gamma(2h_l + p)}{N_{l,p}^2 \Gamma(2h_l)p!},
\tag{17}
$$

when the operators are normalized to have the two-point functions $\langle \mathcal{O}_l(z)\mathcal{O}_l(0)\rangle = z^{-2h_l}$. Ultimately, we want to choose our states to have unit norm. For now, however, we will set $N_{l,p} = 1$ in order to avoid clutter, and put the normalization factors back in at the end. Then, we can replace the external states with integrals over the corresponding operators as follows:

$$
\langle l|\mathcal{O}_R(y_1)\mathcal{O}_R(y_2)|l'\rangle_{\text{cyl}} = \oint \frac{dx}{x} x^{h_l+p} \oint \frac{dz}{z} z^{h_{l'}-p} (y_1\overline{y}_1 y_2\overline{y}_2)^h \langle \mathcal{O}_l(x)\mathcal{O}_R(y_1)\mathcal{O}_R(y_2)\mathcal{O}_{l'}(z)\rangle.
\tag{18}
$$

In practice, the $dx$ and $dz$ contour integrals can readily be evaluated by residues by shrinking them to wrap the poles at $x = \infty$ and $z = 0$ respectively. In finite volume, the interaction term is $\mathcal{O}_R$ integrated over space on the cylinder:

$$
V(t) = V_R(t) - E_{\text{vac}}(g), \quad V_R(t) \equiv \int_0^{2\pi} d\theta \, \mathcal{O}_R(e^{t+i\theta}, e^{t-i\theta}).
\tag{19}
$$

We will suppress $E_{\text{vac}}(g)$ in the next few lines in order to avoid clutter, and restore it shortly. Next, note that the correlator above appears in our expression for $P_{\text{eff}}$ from the Dyson series:

$$
\begin{aligned}
(P_+)_{\text{eff}}|_{g^2} &= \lim_{t \to \infty} \langle l|gV(t)U(t)|l'\rangle_{\text{cyl}}|_{g^2} \\
&= \left(\frac{g}{2\pi}\right)^2 \int_0^{2\pi} d\theta_1 d\theta_2 \langle l|O_R(e^{i\theta_1}) \int_{-\infty}^0 \frac{dt_2}{i} \mathcal{O}_R(e^{t_2+i\theta_2})|l'\rangle_{\text{cyl}},
\end{aligned}
\tag{20}
$$

where we have used the fact that the denominator $U = 1 + \mathcal{O}(g^2)$ does not contribute at this order, and we have used time translation invariance to shift the integration range. We can also use rotational invariant to do one of the angular integrals,

$$
(P_+)_{\text{eff}}|_{g^2} = \frac{g^2}{2\pi} \int_0^{2\pi} d\theta_2 \int_{-\infty}^0 \frac{dt_2}{i} \langle l|O_R(1)\mathcal{O}_R(e^{t_2+i\theta_2})|l'\rangle_{\text{cyl}}.
\tag{21}
$$

Consider the contribution to the expression above from just the disconnected part of the correlator:

$$
\left(\langle l|O_R(y_1)\mathcal{O}_R(y_2)|l'\rangle_{\text{cyl}}\right)_{\text{disc}} \equiv \langle l|l'\rangle \langle O_R(y_1)\mathcal{O}_R(y_2)\rangle_{\text{cyl}}.
\tag{22}
$$

It is straightforward to evaluate this contribution to $P_{\text{eff}}$ by doing the integral directly. However, instead let us first note that by inserting a complete set of states, we can write the contribution from the disconnected piece as follows:[5]

$$
\begin{aligned}
\left(\frac{g}{2\pi}\right)^2 \delta_{ll'} \sum_n \int_0^{2\pi} d\theta_1 d\theta_2 \int_{-\infty}^0 \frac{dt_2}{i} \langle O_R(e^{i\theta_1})|n\rangle \langle n|e^{iE_n t_2} \mathcal{O}_R(e^{i\theta_2})\rangle_{\text{cyl}} &= -\left(\frac{g}{2\pi}\right)^2 \delta_{ll'} \sum_n \frac{|\langle \text{vac}|V|n\rangle|^2}{E_n} \\
&= \delta_{ll'} E_{\text{vac}}|_{g^2}.
\end{aligned}
\tag{23}
$$

---

[5]We define the limit $t \to -\infty$ to be taken in a slightly imaginary direction, so $\lim_{t \to -\infty} e^{it} \equiv \lim_{t \to -\infty(1-i\epsilon)} e^{it} = 0$.

This exactly cancels against the vacuum energy contribution in $V$ that we have been suppressing:

$$\lim_{t\to\infty}\langle l|(-E_{\text{vac}}(g))U(t)|l'\rangle|_{g^2}=-\delta_{ll'}E_{\text{vac}}|_{g^2}\,,\tag{24}$$

since $P_+$ vanishes on the chiral states in the CFT limit $g=0$ and therefore $U(t)=1$ acting on them at this order. Consequently, only the connected part of the correlator contributes to the effective Hamiltonian.

When we pass to the coordinates on the plane, it will be convenient to work with variables $r=e^t$ and $y=e^{i\theta}$ so that we can continue to keep the time and angular integrals separate; in this case, the anti-holomorphic variable $\overline{y}$ is given by $r^2=y\overline{y}$. Using the expression (18),

$$(P_+)_{\text{eff}}|_{g^2}=g^2(2\pi)^2\int_0^1\frac{dr}{r}r^{2h}\oint\frac{dx}{2\pi ix}\frac{dy}{2\pi iy}\frac{dz}{2\pi iz}x^{h_l+p}z^{h_{l'}-p}\langle\mathcal{O}_l(x)\mathcal{O}_R(1)\mathcal{O}_R(y,\overline{y})\mathcal{O}_{l'}(z)\rangle_{\text{conn}}.\tag{25}$$

Next, we focus on the contribution from the most singular terms in the OPE of $\mathcal{O}_l$ and $\mathcal{O}_{l'}$ with the same factor of $\mathcal{O}_R$. For concreteness, it may be helpful for the reader to consider the following argument in the context of the specific case where $\mathcal{O}_l$ and $\mathcal{O}_{l'}$ are both the stress tensor $T$, in which case

$$\frac{\langle T(x)\mathcal{O}(y_1)\mathcal{O}(y_2)T(z)\rangle_{\text{conn}}}{\langle T(\infty)T(0)\rangle\langle\mathcal{O}(y_1)\mathcal{O}(y_2)\rangle}=\frac{2}{c}\left(\frac{h^2(y_1-y_2)^4}{(x-y_1)^2(x-y_2)^2(y_1-z)^2(y_2-z)^2}\right.$$
$$\left.+\frac{2h(y_1-y_2)^2}{(x-y_1)(x-y_2)(y_1-z)(y_2-z)(x-z)^2}\right),\tag{26}$$

where $h$ is the conformal weight of $\mathcal{O}$. In the contour integral in (25), we are free to deform the $x$ and $z$ contours away from their poles at $\infty$ and $0$, respectively, and sum over the other poles. Then there will be terms in this sum over poles where the $x$ and $z$ contour encircle the same pole (e.g., both encircle the point $y_1$) as well as terms where they encircle different poles (one at $y_1$ and the other at $y_2$, say). The former of these contain the double OPE terms where both $\mathcal{O}_l$ and $\mathcal{O}_{l'}$ approach the same factor of $\mathcal{O}_R$. To extract the leading singular terms when $\mathcal{O}_l(x)$ and $\mathcal{O}_{l'}(z)$ approach $\mathcal{O}_R(y_1)$, say, take

$$x=y_1(1+x'),\quad z=y_1(1+z'),\tag{27}$$

and take $x',z'\to 0$ with $x'/z'$ fixed. The leading terms in this limit can be written as

$$\mathcal{O}_l(x)\mathcal{O}_R(y_1)\mathcal{O}_{l'}(z)=\frac{1}{y_1^{h_l+h_{l'}}(x'-z')^{h_l+h_{l'}}}m_{ll'}\left(\frac{z'}{x'}\right)\mathcal{O}_R(y_1)+\dots,\tag{28}$$

for some function $m_{ll'}(u)$. The subleading terms above are suppressed by powers of $x'$ and $z'$. In the case of equation (26), this function is $m_{TT}(u)=\frac{2}{c}(\frac{h^2(1-u)^4}{u^2}+\frac{2h(1-u)^2}{u})$. One convenient property of the function $m_{ll'}(u)$ is that it has a finite Laurent series, because it cannot have a singularity at $x'=0$ greater than $x'^{-2h_l}$ and it cannot have a singularity at $z'=0$ greater than $z'^{-h_{l'}}$. To see how this expansion behaves in the limit of large $p$, write $x$ and $y$ in angular coordinates:

$$x=y_1e^{ix^-/p},\quad z=y_1e^{iz^-/p},\tag{29}$$

where $x^-,z^-$ are integrated from $-p\pi$ to $p\pi$; the reason for choosing this notation is that in the limit of large $p$, we will see how they match onto the infinite volume lightcone coordinates. Taking the expression (25) for $(P_+)_{\text{eff}}$, inserting the operator product expansion (28), changing to $x^-,z^-$ coordinates, and expanding at large $p$, we arrive at the following result:

$$(P_+)_{\text{eff}}|_{g^2}\supset p^{h_l+h_{l'}-2}\left[\int_{-\pi p}^{\pi p}dx^-dz^-\frac{e^{i(x^--z^-)}m_{ll'}\left(\frac{z^-+i\epsilon}{x^--i\epsilon}\right)}{i^{h_l+h_{l'}}(x^--z^--i\epsilon)^{h_l+h_{l'}}}\right]$$
$$\times g^2\int_0^1\frac{dr}{r}r^{2h}\oint\frac{dy}{2\pi iy}\langle\mathcal{O}_R(1)\mathcal{O}_R(y,\overline{y})\rangle.\tag{30}$$

The infinitesimal $\epsilon$ keeps track of the original radial ordering $|x| > |z|$. Now we see that the subleading terms in (28) are suppressed by powers of $p$ due to their additional factors of $x'$ and $z'$. We also recognize the term on the second line as the second order vacuum energy:

$$g^2 \int_0^1 \frac{dr}{r} r^{2h} \oint \frac{dy}{2\pi i y} \langle \mathcal{O}_R(1) \mathcal{O}_R(y, \overline{y}) \rangle = -E_{\text{vac}}|_{g^2} .$$ (31)

Finally, reintroducing the normalization factors $N_{l,p}$ and $N_{l',p}$, we define the matrix elements $M_{ll'}$ in terms of the integral in square brackets:

$$
\begin{aligned}
M_{ll'} &\equiv -\lim_{p\to\infty} 2p \cdot \frac{p^{h_l + h_{l'} - 2}}{N_{l,p} N_{l',p}} \left[ \int_{-\pi p}^{\pi p} dx^- dz^- \frac{e^{i(x^- - z^-)} m_{ll'}\left( \frac{z^- + i\epsilon}{x^- - i\epsilon} \right)}{i^{h_l + h_{l'}} (x^- - z^- - i\epsilon)^{h_l + h_{l'}}} \right] \\
&= -\frac{1}{\pi} \sqrt{\Gamma(2h_l)\Gamma(2h_{l'})} \int_{-\infty}^{\infty} dx^- dz^- \frac{e^{i(x^- - z^-)} m_{ll'}\left( \frac{z^- + i\epsilon}{x^- - i\epsilon} \right)}{i^{h_l + h_{l'}} (x^- - z^- - i\epsilon)^{h_l + h_{l'}}} .
\end{aligned}
$$ (32)

In terms of $M_{ll'}$, we can write this contribution to the $P_{\text{eff}}$ in the following simple form:

$$(P_+)_{\text{eff}}|_{g^2} \supset \frac{M_{ll'}}{2p} (-E_{\text{vac}}|_{g^2}) .$$ (33)

To complete the argument, we need to consider the other poles that contribute to the contour. If both $x$ and $z$ encircle $y_2$ instead of $y_1$, the result is identical to the one we have just derived and simply contributes an extra factor of 2. Crucially, at higher orders $g^n$, there are $n$ such contractions and their sum produces an extra factor of $n$, as described in the previous subsection. Finally, if $x$ and $z$ encircle different poles, in this case $y_1$ and $y_2$ respectively, then we have found by explicit evaluation of the contour integrals, at least in the case of a single relevant operator $\mathcal{O}_R$ with dimension $\Delta < 1$, that the result vanishes at large $p$. The intuitive explanation for this is that when we pass to $x^-$ and $z^-$ variables, $x$ and $z$ are expanded around different points, and the term $(x/z)^p$ in the original integrand will contain the factor $(y_1/y_2)^p$. Since $\oint dy_1$ is an angular integral, this factor can be written as $\sim \int d\theta e^{ip\theta}$ which oscillates rapidly and integrates to zero; in other words, it is a Fourier transform with a large momentum, and without another equal and opposite large momentum somewhere else, it vanishes by momentum conservation. The difficulty in proving this statement more generally is that this argument can fail if there is a nonnegligible contribution from very high spin (order $\mathcal{O}(p)$) descendants of $\mathcal{O}_R$, since their spin can absorb the extra momentum on the circle at finite volume. At second order in $g$, when $\Delta < 1$, it is possible to check by explicit evaluation of the Fourier transforms that the leading contributions at large $p$ come from the OPE poles when both $x$ and $z$ approach the same insertion of $\mathcal{O}_R$. For $\Delta \geq 1$, or for multiple relevant deformations, a more careful analysis is needed beyond the scope of this paper.

In summary, putting it all together, we find

$$(P_+)_{\text{eff}}|_{g^2} = \frac{M_{ll'}}{2p} (-2 \cdot E_{\text{vac}}|_{g^2}) .$$ (34)

## 2.3 Evaluation of matrix elements

In this subsection, we will show how one can evaluate the matrix elements $M_{ll'}$ from the integral (32) in practice. We will also discuss how they match onto an infinite volume formulation of the matrix elements.

By conformal invariance, the four-point function in (25) can always be written as

$$
\langle \mathcal{O}_l(x)\mathcal{O}_R(y_1)\mathcal{O}_R(y_2)\mathcal{O}_{l'}(z)\rangle = \left(\frac{y_1-z}{x-y_1}\right)^{h_l-h_{l'}} \frac{1}{(y_1-y_2)^{2h}(\overline{y}_1-\overline{y}_2)^{2h}(x-z)^{h_l+h_{l'}}}
$$
$$
\times F_{ll'}\left(\frac{x-y_1}{x-y_2}\frac{y_2-z}{y_1-z}\right),
$$
(35)

for some function $F_{ll'}$. Crossing symmetry $y_1 \leftrightarrow y_2$ implies that $F_{ll'}(u)$ satisfies

$$
F_{ll'}(u^{-1}) = u^{h_{l'}-h_l}F_{ll'}(u). \tag{36}
$$

An efficient way to compute the function $F_{ll'}$ is to conformally map $x$ to $\infty$, $z$ to $0$, and $y_1$ to 1, where the correlator above reduces to

$$
\langle \mathcal{O}_l|\mathcal{O}_R(1)\mathcal{O}_R(y)|\mathcal{O}_{l'}\rangle \equiv \langle \mathcal{O}_l(\infty)\mathcal{O}_R(1)\mathcal{O}_R(y)\mathcal{O}_{l'}(0)\rangle = \frac{1}{(1-y)^{2h}(1-\overline{y})^{2h}}F_{ll'}(y). \tag{37}
$$

Note that $|\mathcal{O}_l\rangle$ is the state corresponding to $\mathcal{O}_l$ in radial quantization and should not be confused with the fixed-momentum state $|l\rangle$. For chiral operators $\mathcal{O}_l, \mathcal{O}_{l'}$, the states $|\mathcal{O}_l\rangle$ and $|\mathcal{O}_{l'}\rangle$ can be constructed in terms of the algebra (e.g. if $\mathcal{O}_l = T$, then $|\mathcal{O}_l\rangle = L_{-2}|0\rangle$), so by repeated applications of the chiral algebra one can compute $F_{ll'}(y)$ as a Laurent expansion:

$$
F_{ll'}(y) = \sum_{m=-h_{l'}}^{h_l} a_m^{(ll')}y^m. \tag{38}
$$

One can then substitute $F_{ll'}(y)$ back into (35), and in turn back into the integral (25). Note that, since the external operators are chiral, the four-point function (37) is effectively just a two-point function of the deformation $\mathcal{O}_R$, acted on by the chiral algebra, and therefore (25) is fully determined by the action of the chiral algebra on $\mathcal{O}_R$. In fact, even for finite $p$, it is relatively efficient to do the contour integrals in (25) analytically by taking residues, so one can see in full detail how the large $p$ limit is approached.

Next, we want to read off the function $m_{ll'}(u)$ defined previously in terms of the function $F_{ll'}$. To do this, we can apply the OPE (28) inside the four-point function (35), to find

$$
m_{ll'}(u) = (-u)^{h_l-h_{l'}}F_{ll'}(u^{-1}) = (-1)^{h_l-h_{l'}}F_{ll'}(u). \tag{39}
$$

The matrix elements $M_{ll'}$ can now be determined from the function $F_{ll'}$ by performing the integral in (32). We will do this term-by-term in the expansion of $F_{ll'}(u)$ in powers of $u$, from (38). First, though, it will be instructive to do the finite $p$ integral that gave rise to (32). In particular, (32) was obtained from the large $p$ limit of the following set of contour integrals:

$$
\oint \frac{dx}{2\pi ix}\frac{dz}{2\pi iz}x^{h_l+p}z^{h_{l'}-p}\frac{1}{(x-z)^{h_l+h_{l'}}}m_{ll'}\left(\frac{z-y_1}{x-y_1}\right). \tag{40}
$$

We can change variables to set $y_1 = 1$ without loss of generality. A single term in (38) corresponds to a term $(-1)^{h_l-h_{l'}}u^m$ in $m_{ll'}(u)$, and the corresponding contour integral

$$
\oint \frac{dx}{2\pi ix}\frac{dz}{2\pi iz}x^{h_l+p}z^{h_{l'}-p}\frac{(-1)^{h_l-h_{l'}}(\frac{z-1}{x-1})^m}{(x-z)^{h_l+h_{l'}}}, \tag{41}
$$

can be done in closed form, and one finds that at leading order in $p$, the result is $\frac{p^{h_l+h_{l'}-1}}{\Gamma(h_l+h_{l'})}$. This apparently contradicts the scaling obtained in (30), which was proportional to $p^{h_l+h_{l'}-2}$. However, this discrepancy is resolved after accounting for the fact that the coefficients in the

sum (38) are not all independent of each other, for two reasons. The first is crossing symmetry (36), which implies that $a_m^{(ll')} = a_{h_l - h_{l'} - m}^{(ll')}$, and the second is that the disconnected piece singularity at $x = z$ has been removed, which implies that $F_{ll'}(1) = \sum_m a_m^{(ll')} = 0$. The fact that $\sum_m a_m^{(ll')} = 0$ is the key point here, and it implies that if a contribution from an individual power $u^m$ in $m_{ll'}(u)$ is independent of $m$, then it vanishes after we perform the sum over $m$. To remove such contributions from the very beginning, and also to simultaneously take advantage of crossing symmetry, it is convenient to consider instead a contribution from $m_{ll'}(u)$ of the form

$$\frac{u^m + u^{h_l - h_{l'} - m}}{2} - 1 . \tag{42}$$

Now, the contour integral of interest is

$$I_{h_l, h_{l'}}(m) \equiv - \lim_{p \to \infty} \frac{p}{N_{l,p} N_{l',p}} (2\pi)^2 \oint \frac{dx}{2\pi i x} \frac{dz}{2\pi i z} x^{h_l + p} z^{h_{l'} - p} (-1)^{h_l - h_{l'}} \frac{\frac{(\frac{z-1}{x-1})^m + (\frac{z-1}{x-1})^{h_l - h_{l'} - m}}{2} - 1}{(x - z)^{h_l + h_{l'}}} . \tag{43}$$

This can be evaluated in closed form for $-h_{l'} \leq m \leq h_l$ and the result is

$$I_{h_l, h_{l'}}(m) = \frac{(-1)^{h_l + h_{l'}} 2\pi \sqrt{\Gamma(2h_l)\Gamma(2h_{l'})}}{\Gamma(h_l + h_{l'} - 1)} \left( \frac{|m| + |h_l - h_{l'} - m|}{4} + \text{ind't of } m \right) . \tag{44}$$

The $m$-independent term equals $\frac{h_l - h_{l'}}{4}$, and can be discarded since, as discussed above, it does not contribute once we have removed the disconnected part of the correlator.

Given the expansion coefficients $a_m^{(ll')}$ from (38), the function $I_{h_l, h_{l'}}(m)$ is all we need in practice in order to evaluate the matrix elements $M_{ll'}$:

$$M_{ll'} = (-1)^{h_l + h_{l'}} 2\pi \frac{\sqrt{\Gamma(2h_l)\Gamma(2h_{l'})}}{\Gamma(h_l + h_{l'} - 1)} \sum_{m=0}^{h_l + h_{l'}} a_{m - h_{l'}}^{(ll')} \frac{|m - h_{l'}| + |m - h_l|}{2} . \tag{45}$$

However, we would now like to go farther and understand how to evaluate the matrix elements of the interaction term $\int dx^- \mathcal{O}_R(x^-) \mathcal{O}_R(\infty)$ directly in the infinite volume limit. To do this, let us return to the definition of $M_{ll'}$ from (32), and again consider a single term $(-1)^{h_l - h_{l'}} \frac{1}{2}(u^m + u^{h_l - h_{l'} - m} - 2)$ from $m_{ll'}(u)$. The corresponding integral in (32) is

$$-\frac{1}{\pi} \sqrt{\Gamma(2h_l)\Gamma(2h_{l'})} \int_{-\infty}^{\infty} dx^- dz^- e^{ip(x^- - z^-)} \frac{(-1)^{h_l + h_{l'}} \frac{1}{2} \left( \left( \frac{z^- + i\epsilon}{x^- - i\epsilon} \right)^m + \left( \frac{z^- + i\epsilon}{x^- - i\epsilon} \right)^{h_l - h_{l'} - m} - 2 \right)}{i^{h_l + h_{l'}} (x^- - z^- - i\epsilon)^{h_l + h_{l'}}} . \tag{46}$$

Since $h_l, h_{l'}$ and $m$ are all integers, we can do the $x^-$ and $z^-$ integrals by residues, first closing the $x^-$ contour in the upper half-plane to pick up the poles at $x^- = z^- + i\epsilon$ and $x^- = i\epsilon$, and then closing the $z^-$ contour in the lower half-plane to pick up the remaining residue at $z^- = -i\epsilon$. The result is exactly the same as (44).[6]

---

[6]The exact form of the $m$-independent terms differs, and in fact is sensitive to whether one does the $x^-$ integral or $z^-$ integral by contours first, but such differences do not contribute to the final matrix elements $M_{ll'}$.

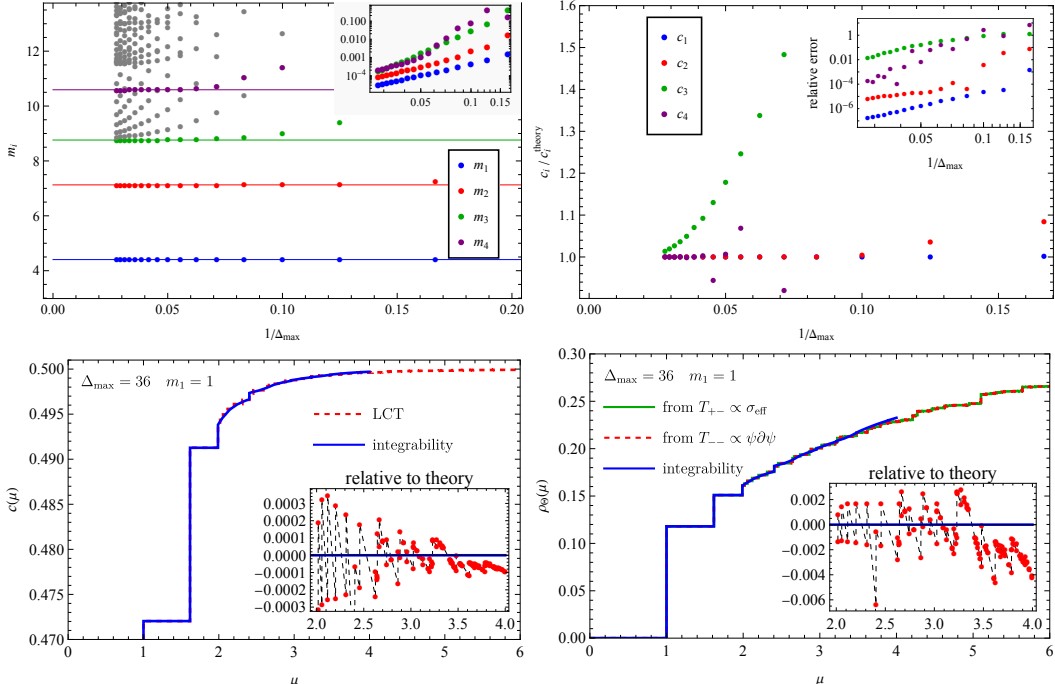

Figure 1: LCT numerical results for the $\sigma$ deformation of Ising, using the lightcone effective action (5). The top-left panel is the mass spectrum as a function of $\Delta_{\max}$, where the points with color are the first 4 bound states which are compared with the integrability results. The top-right panel is the bound states' contribution to Zamolodchikov $c$-function as a function of the truncation $\Delta_{\max}$. The bottom-left panel is the $c$-function. The red dashed line is the LCT result and the blue solid line is the integrability result. The full $c(\mu)$ function and $\rho_\Theta(\mu)$ are computed using $\Delta_{\max} = 36$. The bottom-right panel shows the spectral density of the trace of the stress tensor, $\Theta$. Dashed lines with green and red colors are the LCT numerical results obtained from measuring operators $T_{+-}(x) \propto \sigma_{\text{eff}}(x) = \langle\sigma\rangle\sigma(x)\sigma(\infty)$ and $T_{--} \propto \psi\partial\psi$, respectively, which are related by the Ward Identity. The blue solid line is the integrability result taking only the two-particle form-factors, which is valid up to center of mass energy $\mu \leq 4m_1$.

Table 1: Low lying bound state spectrum and $c$-function contribution. In LCT, we use a basis with truncation $\Delta_{\max} = 36$ including 471 states. The values are compared with exact values from integrability.

| observable | LCT $\Delta_{\max} = 36$ | integrability |
|---|---|---|
| $m_1/(\frac{g}{2\pi})^{8/15}$ | 4.40505 | 4.40491 |
| $m_2/m_1$ | 1.61812 | 1.61803 |
| $m_3/m_1$ | 1.98932 | 1.98904 |
| $c_1$ | 0.47203836 | 0.47203828 |
| $c_2$ | 0.01923113 | 0.01923126 |
| $c_3$ | 0.00259197 | 0.00255724 |

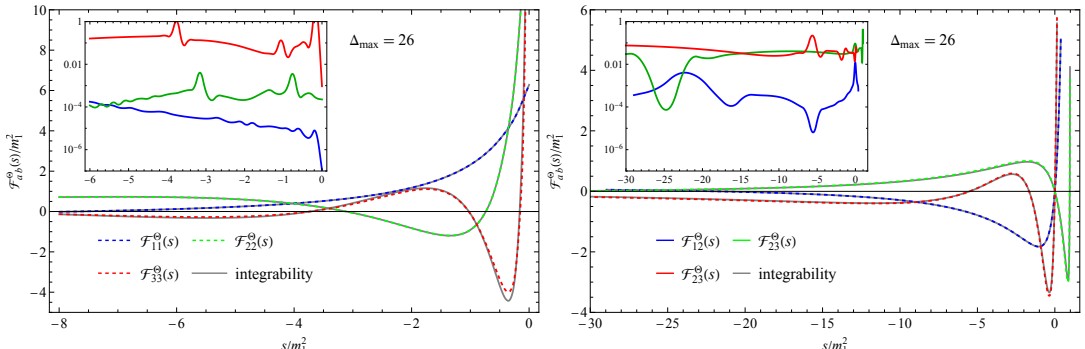

Figure 2: Form factor $\mathcal{F}^{\Theta}_{a,b}(s) = \langle a, p_1 | \Theta(0) | b, p_2 \rangle$ in Ising CFT deformed by $\sigma$ operator, for the lowest 3 bound states $a, b = 1, 2,$ and 3. We use variable $s = -(p_1 + p_2)^2$. The integrability results are shown in gray lines to compare with the numerical truncation results in blue, green, and red colors. The inset shows the relative error between each numerical truncation result and integrability result in log frame.

## 3 Comparison of lightcone effective interaction and integrability results

The most important test of our formalism is to check whether our infinite $p$ lightcone effective Hamiltonian (5), which we reproduce here

$$(P_+)_{\text{eff}} = \frac{m^2}{2} \int dx^- \psi \frac{1}{i\partial_-} \psi(x^-) + g \langle \sigma \rangle \int dx^- (\sigma(x)\sigma(\infty) - \mathbb{1}), \tag{5}$$

actually gives us the right answer for physical observables. We take the integrable limit $m = 0$ where these observables are known exactly. To efficiently evaluate these coefficients, we utilize an optimal basis for the fermion mode expansion in the four point function $\langle \mathcal{O}_l(x)\sigma(y_1)\sigma(y_2)\mathcal{O}_{l'}(z) \rangle$. For details, see Appendix B. The algorithm allows us to compute $P_+^{\text{eff}}$ up to $\Delta_{\max} = 36$ (which has 471 states) in $\sim 10$ minutes. The code to compute the Hamiltonian matrix elements is publicly available.[7]

**Spectrum**

In Figure 1, we show the comparison between various observables computed from LCT and those computed from integrability. The most straightforward observable is the spectrum. We diagonalize the effective Hamiltonian and extract the eigenvalues corresponding to the beginning 4 bound states, $m_i$, $i = 1, .., 4$. The first 3 bound states are below the 2-particle continuum and are easy to isolate. The 4th bound state is chosen to be the state whose eigenvalue is the closest to the integrability result, and we see later that this choice also gives a consistent value for the contribution to the central charge, $c_4$. At $\Delta_{\max} = 36$ the mass eigenvalues are all correct to about 0.01%. The numerical values computed using our highest truncation is quoted in Table 1. We have used the exact result for $\langle \sigma \rangle$ [10, 11]

$$\langle \sigma \rangle = g^{\frac{1}{15}} \frac{8\Gamma(\frac{1}{5})\Gamma(\frac{1}{3})\Gamma(\frac{7}{15})}{15\pi\Gamma(\frac{8}{15})\Gamma(\frac{2}{3})\Gamma(\frac{4}{5})} \left( \frac{\Gamma(\frac{3}{4})\Gamma^2(\frac{13}{16})}{\Gamma(\frac{1}{4})\Gamma^2(\frac{3}{16})} \right)^{\frac{8}{15}} = 0.1798858566 g^{\frac{1}{15}}. \tag{47}$$

---

[7]https://github.com/andrewliamfitz/LCT.

The exact result for the lightest mass $m_1$ is [10, 11]

$$m_1^2 = g\langle\sigma\rangle \times \left(30\sin\left(\frac{\pi}{3}\right)\sin\left(\frac{\pi}{5}\right)\cos\left(\frac{\pi}{30}\right)\right) = 2.732007827 g^{\frac{16}{15}}. \tag{48}$$

Remarkably, even using only a single state $|T, p\rangle$, created by the stress tensor, one obtains an approximate prediction for $\frac{m_1^2}{g\langle\sigma\rangle} \approx M_{TT} = \frac{45\pi}{8} = 17.67$, which differs from the exact answer $30\sin(\frac{\pi}{3})\sin(\frac{\pi}{5})\cos(\frac{\pi}{30}) = 15.19$ by only 16%.

**Spectral function**

Next we compute the Zamolodchikov $c$-function

$$c(\mu) = \int_0^{\mu^2/2} dp_+ |\langle p_+|T_{--}|0\rangle|^2, \quad T_{--} = \frac{1}{2}\psi\partial_-\psi. \tag{49}$$

The bound states' contribution to the $c$-function

$$c_i = |\langle b_i|T_{--}|0\rangle|^2, \tag{50}$$

have relative errors of $10^{-6}$, $10^{-5}$, $0.01$ and $10^{-4}$ for $i = 1, 2, 3,$ and $4$. The continuum's contribution to the $c$-function is a smooth function, and in LCT is discretized by the finite truncation. In Figure 1, we see that the discrete contributions to the $c$-function agree with the integrability results well to better than 0.03%. The $c$-function and the spectral density of the trace of the stress tensor $T_{+-}$ are related by the Ward identity

$$[P_-, T_{+-}] + [(P_+)_{\text{eff}}, T_{--}] = 0, \tag{51}$$

where $T_{+-}$ as an effective operator is given by

$$T_{+-} = (2 - \Delta_\sigma)g\sigma_{\text{eff}}, \quad \sigma_{\text{eff}}(x) = \langle\sigma\rangle\sigma(x)\sigma(\infty). \tag{52}$$

Here, the dependence of $T_{+-}$ on $\sigma_{\text{eff}}$ is the same as in the analogous equal-time formula [12] with $\sigma$. $\sigma_{\text{eff}}$ is consistent with its integral over $x^-$ being $P_+^{\text{eff}}$. In [2], we observe that $|T_{--}\rangle$ is the same as $|T_{--}^{\text{eff}}\rangle$, the effective operator, because by construction they both measure the momentum $\int dx^- T_{--} = P_-$ and any nonvanishing 'renormalization' of $|T_{--}^{\text{eff}}\rangle$ would lead to inconsistencies. We check the Ward identity (51) by directly computing the spectral density of the $\sigma_{\text{eff}}$ operator and find exact agreement. We include the numerical values of $c_i$ in Table 1.

**Form factors**

To compute the form factors of a primary operator $\mathcal{O}$ in terms of the eigenstates of our truncation, we follow the procedure from [13], which we briefly review here. We define the form factor as the following matrix element,

$$F_{i,j}^{\mathcal{O}}(\theta_i, \theta_j) \equiv \langle\mu_i, p_{i-}|\mathcal{O}(0)|\mu_i, p_{j-}\rangle, \quad p_{i-} \equiv \mu_i e^{-\theta_i}, \tag{53}$$

where $\theta_i, \theta_j$ are the rapidities of the incoming and outgoing particles with masses $\mu_i$ and $\mu_j$, respectively. If the operator $\mathcal{O}$ has spin $\ell$, then it is more convenient to define the following rescaled form factor which is invariant under boosts:

$$\mathcal{F}_{i,j}^{\mathcal{O}}(\theta) \equiv e^{\frac{\ell}{2}(\theta_i+\theta_j)}F_{i,j}^{\mathcal{O}}(\theta_i, \theta_j), \quad \theta \equiv \theta_i - \theta_j. \tag{54}$$

With this definition, the form factors for $\Theta \equiv T^\mu_\mu = 2T_{+-}$ and $T_{--}$ are related by

$$\mathcal{F}^{T_{--}}_{i,j}(\theta) = \left( \frac{e^\theta \mu_j - \mu_i}{e^\theta \mu_i - \mu_j} \right) \mathcal{F}^\Theta_{i,j}(\theta), \tag{55}$$

due to the Ward identity. In particular, note that $\mathcal{F}^\Theta_{i,i} = \mathcal{F}^{T_{--}}_{i,i}$.

After diagonalizing $(P_+)_{\text{eff}}$, we obtain stable one-particle states as linear combinations of basis states:

$$|\mu_i, p_-\rangle = \sum_k c^{(i)}_k |\mathcal{O}_k, p_-\rangle. \tag{56}$$

Inserting this sum into the definition of the form factor, we obtain a sum of the form

$$\mathcal{F}^\mathcal{O}_{i,j}(\theta) = \sum_{k,k'} c^{(i)*}_k c^{(j)}_{k'} \langle \mathcal{O}_k, p_{j-} | \mathcal{O}(0) | \mathcal{O}_{k'}, p_{j'-} \rangle, \tag{57}$$

and each term in this sum is the Fourier transform of a three-point function of primary operator. Such three-point functions are completely determined by conformal invariance, up to an overall coefficient which is the OPE coefficient $C_{\mathcal{O}jj'}$. The Fourier transforms can all be done in closed form, and the final result for the form factor is

$$\mathcal{F}^\mathcal{O}_{i,j}(\theta) = \sum_{k,k'} c^{(i)*}_k c^{(j)}_{k'} C_{\mathcal{O}kk'} e^{(\frac{h_\mathcal{O}}{2} - h_k)\theta} \frac{4\pi(\mu_i \mu_j)^{\frac{h_\mathcal{O}}{2}}}{\Gamma(h_k + h_{k'} + h_\mathcal{O} - 1)} \sqrt{\Gamma(2h_k)\Gamma(2h_{k'})} P^{(2h_k - 1, 1 - 2h_\mathcal{O})}_{h_\mathcal{O} + h_{k'} - h_k - 1}(1 - 2\frac{\mu_i}{\mu_j} e^{-\theta}), \tag{58}$$

where $P^{(\alpha,\beta)}_n$ is a Jacobi polynomial, defined to vanish when $n < 0$. As the truncation parameter $\Delta_{\max}$ is increased, this expression for the form factor converges uniformly for positive real $\theta$, but its derivative does not converge due to increasingly rapidly oscillatory errors (but with decreasing amplitudes). These rapidly oscillatory errors can be removed to increase the precision of the result where we analytically separate out the positive frequency ($\sim e^{+in\phi}, n > 0$) and negative frequency ($\sim e^{+in\phi}, n < 0$) modes in the variable $\phi$ defined by $1 - 2e^{-\theta} \equiv \cos\phi$ and extend their ranges of convergence using Padé approximations in a well-chosen variable. The details of this improvement procedure are described in [13] (section 4.3.2).

In Fig. 2, we show the result for the computation of the two particle form factors of the trace of the stress tensor $\mathcal{F}^\Theta_{a,b}(s) = \langle a, p_1 | \Theta(0) | b, p_2 \rangle$ for bound states $a, b = 1, 2, 3$ using the method just described. The form factors can also be computed exactly with integrability. At truncation $\Delta_{\max} = 26$, the 1-1, 2-2, and 3-3 diagonal form factors are accurate to $10^{-4}$, $10^{-3}$, and 0.1, respectively, for a large range of particle momenta.

Because of the high accuracy of the form factor $\mathcal{F}^\Theta_{1,1}(s)$, we can reasonably try to analytically continue beyond the physical regime $s < 0$ to $s > 0$. We expect on general grounds to see poles at $s = m_i^2$ for the masses of stable particles, $m_i^2 < 4m_1^2$. To do the extrapolation, we fit $\mathcal{F}^\Theta_{1,1}(s)$ to a rational function, which we choose to be of the form $\frac{P_6(s)}{Q_6(s)}$ with $P_6(s), Q_6(s)$ both sixth-order polynomials whose coefficients are determined by fitting in the regime $s < 0$. The result of this fit is shown for the case $\eta = 0$ in Fig. 3, where we do indeed see a pole near $s = m_1^2$ and $m_2^2$. The pole at $m_3^2 = 3.96m_1^2$ however is too close to the two-particle threshold for this method to work accurately, since starting at $4m_1^2$ there is a branch cut in $s$, beyond which point a rational function of $s$ is not expected to be a good approximation.

The exact position and residues of the poles near $s = m_1^2$ and $m_2^2$ from fitting to a rational function are numerically

$$\mathcal{F}^\Theta_{1,1}(s) \sim -\frac{1.503}{s - 0.9995m_1^2} + \frac{1.17}{s - 2.66m_1^2} \qquad (\Delta_{\max} = 26). \tag{59}$$

where $\sim$ indicates that we only write the pole terms. This is very close to the exact result:

$$\mathcal{F}^\Theta_{1,1}(s) \sim -\frac{1.50626}{s - m_1^2} + \frac{1.13909}{s - 2.62m_1^2} \qquad (\text{exact}). \tag{60}$$

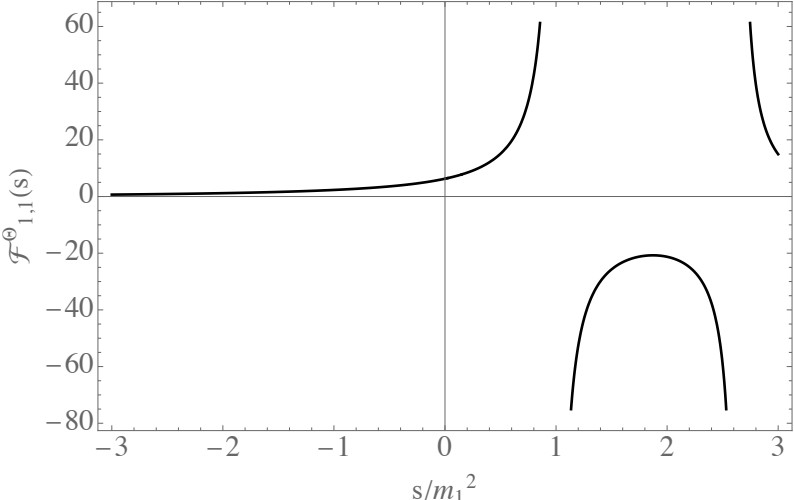

Figure 3: Form factor $F^{\Theta}_{1,1}(s) = \langle 1, p_1 | T(0) | 1, p_2 \rangle$ in IFT at $\eta = 0$, plotted as a function of $s = -(p_1 + p_2)$, continued into the $s > 0$ regime by fitting the $s < 0$ regime to a rational function. There are clearly visible poles near $m_1^2$ and $m_2^2$; the comparison of the position and residues of these poles to the exact results from integrability are given in the text.

## 4  Ising field theory low-temperature phase

So far, we have only considered the integrable limit of Ising Field Theory where the $\epsilon$ deformation is absent. In this section, we will apply the effective Hamiltonian (5) to the full low-temperature phase, with both deformations turned on. The fermion mass term has an IR divergence that needs to be either eliminated by choosing a different basis, or regulated by introducing a cutoff. We explain our IR regulator scheme in appendix D. At $g = 0$, $m < 0$, the fermions can be identified as domain walls separating the two degenerate vacua. Adding a small coupling $g$ creates a confining potential between the fermions. In the $g \to 0$ limit, this results in an infinite tower of "mesons" whose spectrum was obtained first by [14] analytically. At finite $g$, these bound states subsequently cross the multiparticle threshold and become resonances. The spectrum of the bound states and resonances can be computed approximately using Truncated Free-Fermion Space Approach (TFFSA) [11] and using Bethe-Salpeter equation [5]. Remarkably, the Bethe-Salpeter analysis remains good all the way to the vicinity of $m = 0$ and some of the resonance masses are close to the $E_8$ theory exact bound state masses and higher thresholds. We will reproduce the spectrum as a check of the LCT effective Hamiltonian, and compute more observables such as the $c$-function and various form factors.

We begin by computing the spectrum at generic $\eta < 0$. The $\langle \sigma \rangle$ is computed to high precision in [11]. We use that result as an input and compute the spectrum for various $\eta$, shown in Figure 4. We take truncation levels up to $\Delta_{\max} = 36$ (417 states) and scan $\eta$, and plot the energy eigenvalues normalized by the first bound state mass $m_1$ in the upper-left panel. In this plot we see a clear separation between the bound states below the $2m_1$ threshold and the multiparticle states above it. At finite and large $|\eta|$ we see many bound states and we can keep track of the change of individual eigenvalue as functions of $\eta$ even across the threshold. This indicates the resonances have long lifetime and can be well approximated by the eigenstates of the truncated Hamiltonian. At small $|\eta|$, we have fewer bound states. We can still qualitatively identify the remnants of the resonances from the eigenvalues, but we see that the eigenvalues strongly repel each other and the identification is only approximate. We also compute the spectrum at fixed values of $\eta$ and track the convergence across a range of truncation $\Delta_{\max}$,

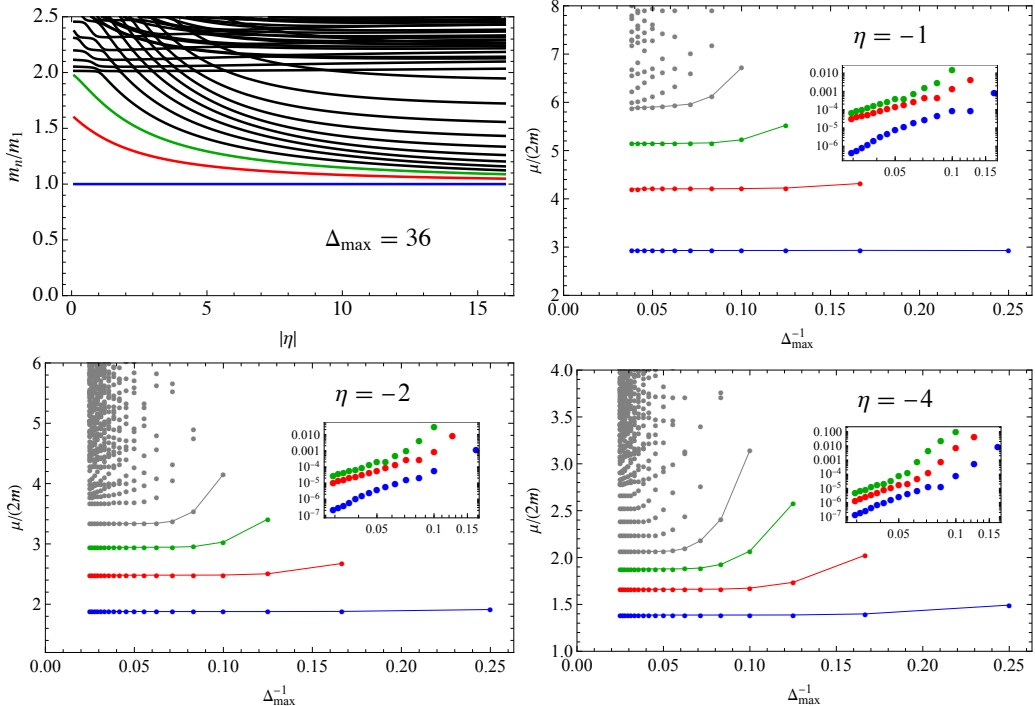

Figure 4: The mass spectrum of IFT in the low temperature phase. We take the truncation up to $\Delta_{\max} = 36$. The inset shows the relative error compared with $\Delta_{\max} \to \infty$ extrapolation, using a model of $m_i(\Delta_{\max}) = a\Delta_{\max}^{-b} + c$.

shown in the other 3 panels. At each $\eta$, there is a number of stable bound states between $m_1$ and $2m_1$. When $\mu \geq 2m_1$, we have a dense set of states forming a quasi-continuum, which represent the multi-particle states. At all $\eta$ values attempted, the bound state masses seem to converge as a high power of $\frac{1}{\Delta_{\max}}$. This suggests it is promising to scale up the truncation and obtain much more accurate numerical results. In Figure 9 we compute the form factors $\mathcal{F}_{a,b}^{\Theta}(s) = \langle a, p_1 | \Theta(0) | b, p_2 \rangle$ between the stress tensor and bound states in general $\eta$ using the same procedure discussed in the previous section. At larger $\eta$ more bound states are allowed, and we show one example form factor for a low bound state and one example for a high bound state.

To see that we get the right IFT spectrum, we compare the numerical spectrum obtained by a low truncation LCT effective Hamiltonian against the TFFSA data in Figure 5. On the TFFSA side, we used data presented in [11], where the basis states are selected in the rest frame with maximum total absolute momentum of fermion creation operator $\Lambda = 20$ in units of $2\pi/R$, which corresponds to 487 states. According to [11], going to the next level $\Lambda = 24$ does not lead to a visible difference.[8] On the LCT side, we take the truncation $\Delta_{\max} = 14$, corresponding to just 16 states, and there is no visible difference between this truncation and our largest truncation $\Delta_{\max} = 36$ (471 states) for the lowest 3 bound state masses, except for the small region where $m_3$ crosses the $2m_1$ threshold. The LCT and TFFSA data agree to better accuracy than what is visible in the plot.

The LCT effective Hamiltonian is not only efficient but also stable across a wide range of parameters. In standard TCSA and TFFSA, the observables depend on the volume, and infinite volume data always requires some extrapolation. Since truncation effects are more severe at large volume, one needs to restrict to a window of volume and truncation. In [2, 15] we

---

[8]All eigenvalues from [11] have finite size errors. For example, [11] shows that the finite size error of the 3 stable particle masses at small $|m|$ begin to exponentially decay at approximately $Rh^{\frac{8}{15}} \gtrsim 5$.

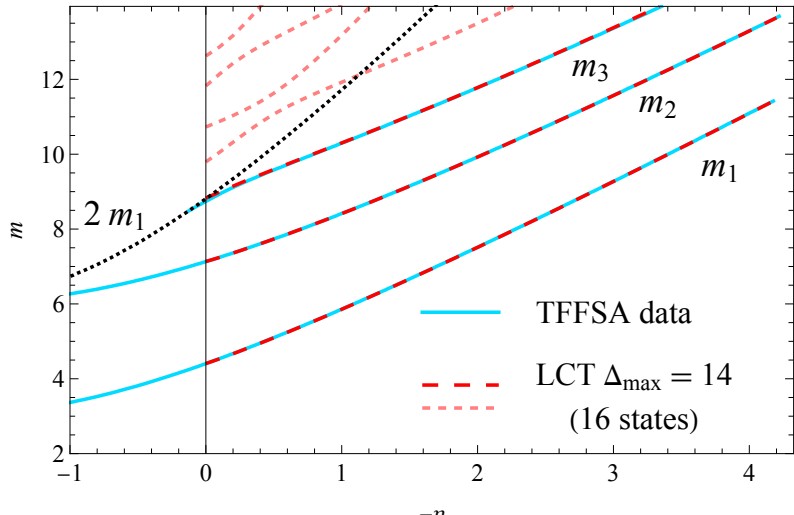

Figure 5: Comparing the spectrum obtained with LCT effective Hamiltonian (4) with the TFFSA numerical data in [11]. The TFFSA data comprises 3 bound state masses $m_1$, $m_2$, and $m_3$ for both positive and negative $\eta$. The LCT data includes 3 bound state masses in the low-T phase $\eta \leq 0$ only, and a few higher eigenvalues are shown as well. We have taken units where $g = 2\pi$.

showed that TCSA can be enhanced by a boost which mitigates the finite volume effects, but at finite momentum $p$, the volume range accessible is still restricted. The window of truncation parameters optimized for different observables can be different for different $\eta$, and the analysis has to be case-by-case. In contrast, the LCT effective Hamiltonian is volume independent, with the truncation $\Delta_{\max}$ being the only parameter controlling the accuracy. In Figure 6 we show a few bound state masses and their contribution to the $c$-function for a wide range $\eta$. The LCT numerical results have all converged with errors smaller than the thickness of the lines. As a comparison, we show the TCSA data at different truncation levels and spacial volumes. The setup with $p = 14$ and $\Lambda = 42$ has 43735 states, and the setup with $p = 30$ and $\Lambda = 36$ has 31266 states. The TCSA result still visibly varies with these unphysical parameters, with large $\eta$ and small $\eta$ requiring different parameters to reach good accuracy. In contrast, LCT with only 471 states has already converged, producing a line smoothly interpolating between TCSA results in different patches of the parameter space.

Without integrability, we no longer know the exact non-peturbative matching between the lightcone and equal-time Hamiltonian, which is controlled by $\langle\sigma\rangle$. If we take an "experimentalist's approach", we just need to fix $\langle\sigma\rangle$ by measuring a physical observable, and the rest of the observables can be compared between lightcone equal-time in terms of dimensionless ratios. In Figure 7 we show various observables as functions of mass ratio $m_2/m_1$. We see that once the underlying parameters for LCT and TCSA are chosen to produce the same $m_2/m_1$, they also agree on other dimensionless observables such as bound states' mass ratios $m_i/m_1$ and their contribution to the central charge $c_i$.

The bound state spectrum in the limit $g \to 0$, approached from the low-temperature phase, is known analytically in [14], providing a precision check of the LCT Hamiltonian (5) in the low temperature phase. The prediction is that the fermions feel a linear confining potential and form infinitely many mesonic states whose masses are given by

$$M_n - 2m = (2\langle\sigma\rangle h)^{\frac{2}{3}} z_n, \tag{61}$$

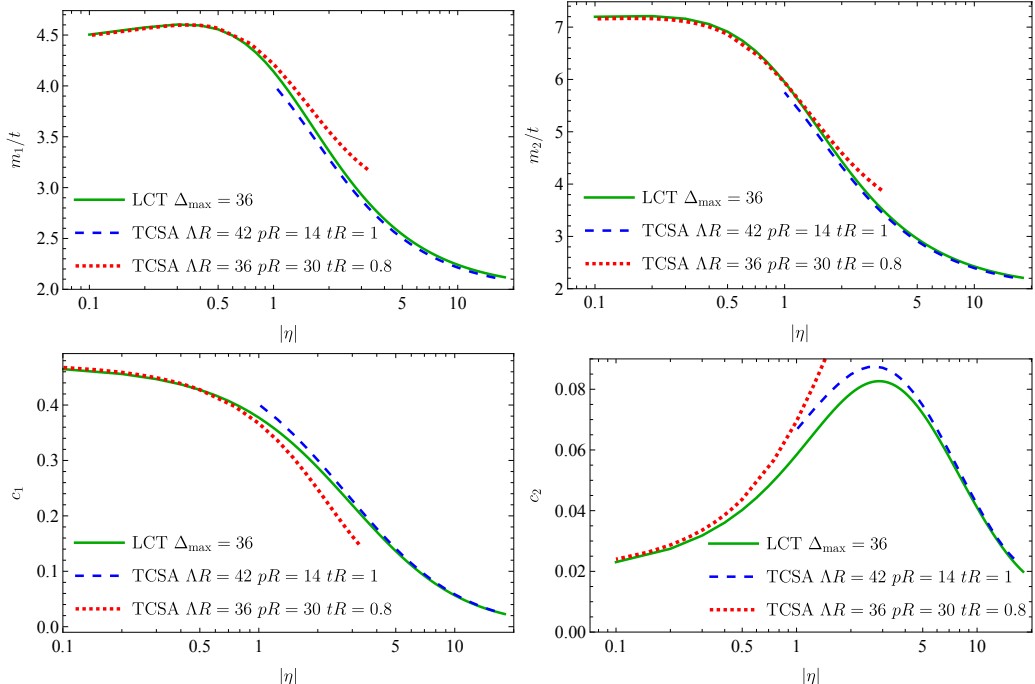

Figure 6: Prediction of the IFT Hamiltonian in LCT with both $\sigma$ and $\epsilon$ deformations in the low temperature phase $\eta < 0$. The $\sigma$ matrix is computed with (4) and we use the numerical result of [11] as the an input for $\langle\sigma(R)\rangle$. We scan $\eta$ and plot the bound state masses in units of $t = \sqrt{m^2 + h^{\frac{16}{15}}}$ and the bound states' contribution to the Zamolodchikov c-function $c_i$. As a comparison, we plot the numerical results of the boosted equal time Hamiltonian truncation. In each plot, we see that the two regimes $|\eta| \lesssim 1$ and $|\eta| \gtrsim 1$ prefer different truncation scheme parameters such as the volume and the boost, whereas the LCT results agrees with equal time results in both regimes, and smoothly interpolates between the two in a region where neither scheme is trustworthy.

where $h = g/(2\pi)$, $z_n$ are roots of the Airy function $\text{Ai}(-z)$, with the vacuum expectation value

$$\langle\sigma\rangle = m^{\frac{1}{8}} 2^{\frac{1}{12}} e^{-\frac{3}{2}\zeta'(-1)} = 1.35783834170660\cdots, \tag{62}$$

used as an input in (5). In Figure 8 we show that the low-lying bound state spectrum from the LCT Hamiltonian (5) matches the theoretical result (61), up to truncation error and finite $h$ corrections.

Above the multi-particle threshold, we no longer have stable bound states for generic $\eta$, and the resonances are "smeared" over multiple eigenvalues. For the purpose of extracting the resonance spectrum, an eigenvalue plot like the upper-left panel of Figure 4 is too noisy. Instead, we plot the spectral density $\rho_\Theta(\mu)$ in Figure 10. At finite truncation we always have a discrete spectrum, rendering $\rho_\Theta(\mu)$ as a sum of isolated delta functions. In order to restore the smooth function, we take an average kernel

$$\rho_\Theta^{\text{smooth}}(\mu) \equiv \sum_i \frac{1}{\pi} \text{Im} \frac{|\langle T_{+-}|\mu_i\rangle|^2}{(\mu - i\epsilon)^2 - \mu_i^2}, \tag{63}$$

with the average window $\epsilon = 0.1 g^{8/15}$. In the large $|\eta|$ regime we see bright isolated spectral lines. Those are sharp resonances that are almost captured by a single eigenstate. When $|\eta|$ is smaller, more eigenstates are involved but we still find the center of the bright region agree

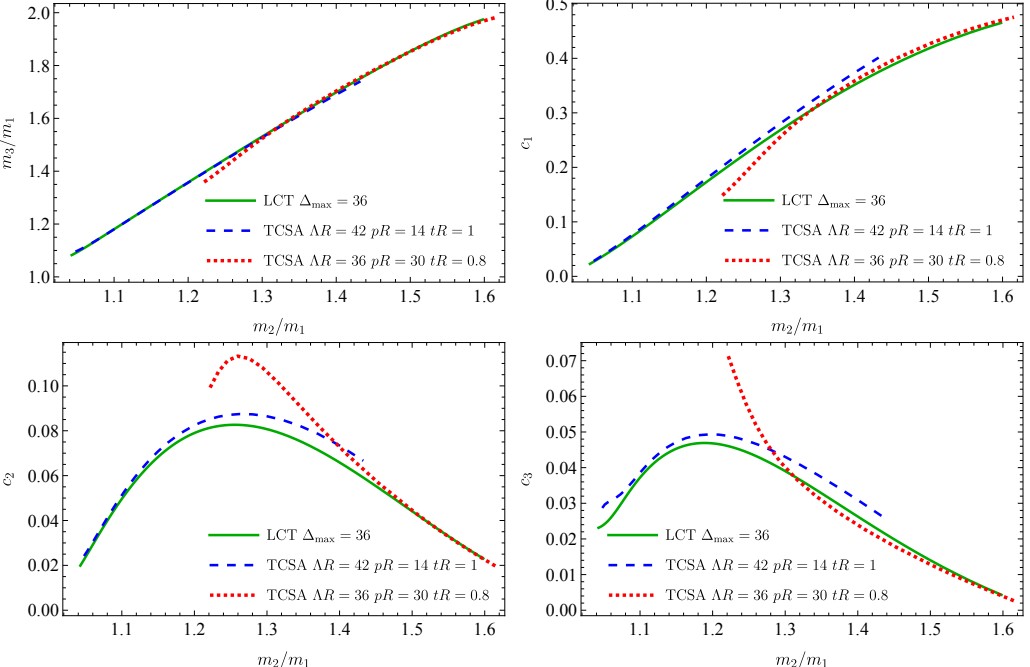

Figure 7: Predictions from lightcone effective Hamiltonian of IFT in the low-temperature phase without the $\sigma$ vev as an input. We scan over the $\sigma$ vev and plot the bound state mass ratio $m_3/m_1$ and bound states' contribution to the Zamolodchikov $c$-function $c_i$ as functions of the ratio $m_2/m_1$, which is shown as the green line. The points represent the TCSA data computed from the large volume equal-time Hamiltonian at different $\eta$. The blue line approximately pass through all points, meaning all predictions of the lightcone effective Hamiltonian are the same as the equal time Hamiltonian when $m_2/m_1$ from both Hamiltonians are tuned to be the same.

with the Bethe-Salpeter analysis in [5]. Near $\eta = 0$ the plot becomes blurred as the resonances' lifetimes become very short.

# Acknowledgments

We are grateful to João Penedones, Matthew Strassler, Hao-Lan Xu, and Xi Yin for helpful discussions, and to Hongbin Chen for collaboration in the early stages of the project.

**Funding information** ALF and EK are supported by the US Department of Energy Office of Science under Award Number DE-SC0015845, and the Simons Collaboration on the Non-Perturbative Bootstrap. YX is supported by a Yale Mossman Prize Fellowship in Physics and by the Simons Collaboration on Confinement and QCD Strings.

# A  Demonstration code

In this appendix we will walk through the practical details of Lightcone Conformal Truncation and explicitly establish a simple code to compute the Ising Field Theory effective Hamiltonian in Mathematica.

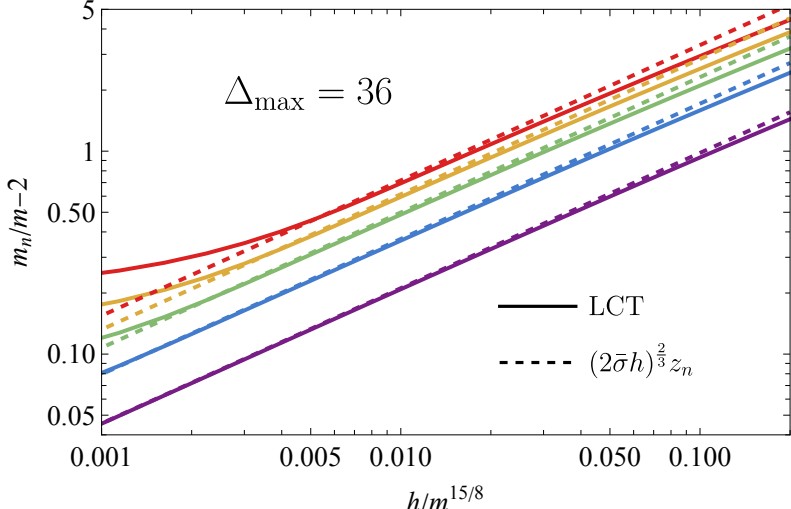

Figure 8: The prediction of the bound states' spectrum in the limit $h \to 0$. As predicted by [5, 14], there is an accumulation point of two fermion bound states at $m_n \to 2m$, and the masses of each bound states is predicted analytically in (61). We use the LCT effective Hamiltonian (5) where the $\langle\sigma(R)\rangle$ input is known analytically from (62). The LCT prediction agrees with the analytic result at small but finite $h$. If $h$ is too small, then the mass separation is comparable with the LCT truncation error. If $h$ is too large, then the two particle approximation gets sizable finite $h$ corrections.

## A.1 The Virasoro minimal basis

We begin by constructing the basis of the effective Hamiltonian. The states (3) are built on chiral quasi-primary operators of 2D Ising CFT. The orthonormal condition of the Fourier transformation is identical to that of the quasi-primary operators up to a $\Delta$-dependent normalization factor

$$\langle l, p | l', p' \rangle = \frac{2\pi e^{i\pi\Delta} p^{2\Delta-1}}{\Gamma(2\Delta)} \delta(p - p') \langle \mathcal{O}_l(0) | \mathcal{O}_{l'}(\infty) \rangle, \tag{A.1}$$

where $\langle \mathcal{O}_l(0) | \mathcal{O}_{l'}(\infty) \rangle$ is the Gram matrix of the CFT states in radial quantization. For now we can forget about this normalization and just divide by it in the final answer, and our goal is to find a basis of radial quantization states. The conceptually simplest way to iterate all such states is to use the chiral Virasoro algebra. We define a "monomial" state to be a state generated by a string of Virasoro operators

$$|L_{-S}\rangle \equiv \prod_{i=1}^{\text{len}(S)} L_{-s_i} |0\rangle \quad S = \{s_1, s_2, \cdots, s_{\text{len}(S)}\}, \tag{A.2}$$

and we take the convention $s_i \geq s_{i+1}$ for all of our monomial states. We can then compute the inner product of the monomial states using the commutator of Virasoro generators, the code to realize it is the following:

```
In[1]:= cIsing = 1/2;
        inner[{_?Negative,z___}]=0;
        inner[{x___,_?Positive}]=0;
        inner[{}]=1;
        inner[{x___,y1_,y2_,z___}]/;y1>y2:=inner[{x,y1,y2,z}]=
        inner[{x,y2,y1,z}]+If[
          y1+y2==0,(2y1*(-Plus[z])+cIsing/12 (y1^3-y1))inner[{x,z}],
          (y1-y2)*inner[{x,(y1+y2),z}]
        ];
```

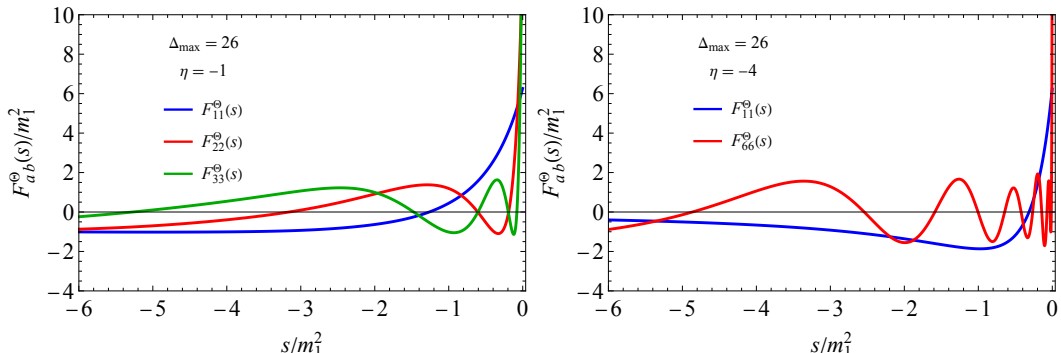

Figure 9: Form factor $\mathcal{F}^{\Theta}_{a,b}(s) = \langle a, p_1 | \Theta(0) | b, p_2 \rangle$ in IFT at $\eta = -1$ and $\eta = -4$, plotted as functions of $s = -(p_1 + p_2)$. At $\eta = -1$, the spectrum has 3 stable bound states belwo the multi-particle threshold, and we show the diagonal form factors of all of them. At $\eta = -4$, the spectrum has many stable bound states, and we show the 1st and 6th bound states for example.

The $c = 1/2$ chiral algebra has a lot of null states. The computation of the Gram matrix and orthogonalization will speed up drastically if we find a minimal basis of the monomial states that are linearly independent. We find empirically in the case of Ising CFT, a procedure of selecting a subset of Virasoro characters based on the partition function always gives us the correct set. The partition function of Ising CFT partition function can be written as the scalar product of Virasoro characters

$$Z_{\text{Ising}} = \left| \chi_{1,1} \right|^2 + \left| \chi_{1,2} \right|^2 + \left| \chi_{2,1} \right|^2 , \tag{A.3}$$

where $\chi_{r,s}$ is the Virasoro character of the primaries $\phi_{1,1} = \mathbb{1}$, $\phi_{1,2} = \epsilon$ and $\phi_{2,1} = \sigma$. The Virasoro characters can be expressed as a sum of Verma modules

$$\chi_{r,s} = \frac{q^{\frac{1}{24}}}{\eta(\tau)} \sum_n q^{\frac{(2np(p+1)+pr-(p+1)s)^2}{4p(p+1)} - \frac{1}{24}} - q^{\frac{(2np(p+1)+pr+(p+1)s)^2}{4p(p+1)} - \frac{1}{24}} . \tag{A.4}$$

If we expand $\chi_{r,s}$ in powers of $q$, the coefficient of each level is the number of states in that level. If we first multiply by $(1-q)$, we will obtain a $q$-series where each coefficient represents the number of quasi-primaries at that level. We observe that at all levels that we have checked, the characters of Ising CFT factorize into the following form

$$\chi_{r,s} = \frac{q^{h_{r,s} - \frac{c-1}{24}}}{\eta(\tau)} (1 - q^{\ell_1})(1 - q^{\ell_2})(1 - q^{\ell_3}) \cdots . \tag{A.5}$$

This is a very suggestive form. It means that we can construct the minimal basis by removing the levels $\ell_i$, and the remaining set

$$\{L_{-S} | \forall i, \, s_i \notin \{\ell_1, \ell_2 \cdots\}\} , \tag{A.6}$$

automatically gives just the correct number of states as well as the number of quasi-primaries at each level determined by the character. At all levels that we checked, the Gram matrix restricted to the minimal basis of monomials (A.6) has full rank, i.e. they are linearly independent. In practice, we can find the Virasoro generators to be removed in the vacuum character using the following code:

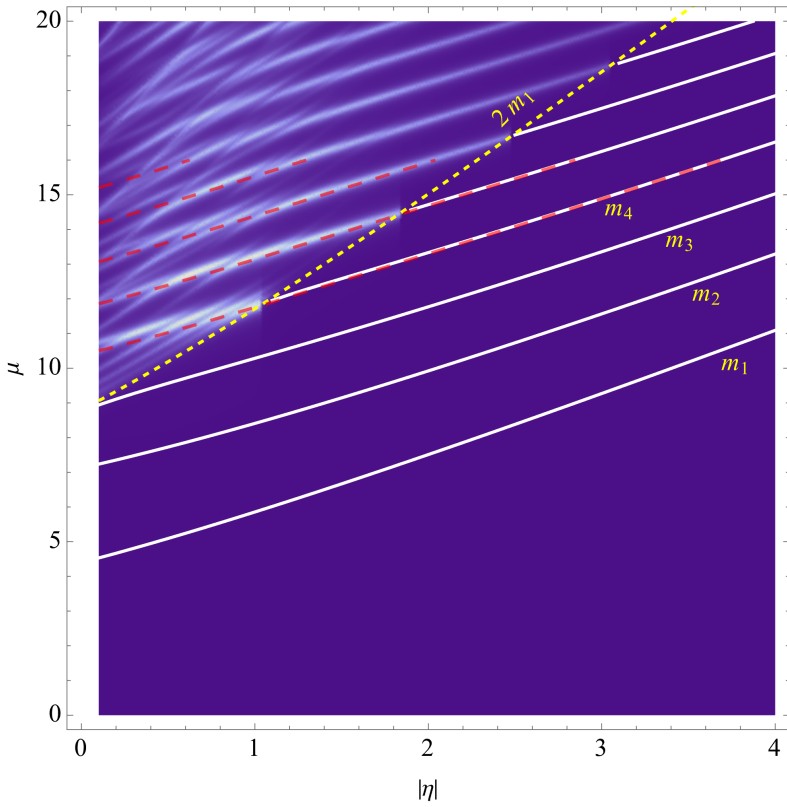

Figure 10: A plot of the (scaled) spectral density of the trace of the stress tensor, $\rho_\Theta(\mu)(\eta^2 + 1)^{-2}$ as a function of both $\mu$ and $\eta$. The truncation is $\Delta_{\max} = 36$ and the units of the plot are set by $g = 1$. To resolve the discrete spectrum, the plot above the $2m_1$ threshold (yellow dashed line) is coarse-grained vertically with an average window of $\epsilon = 0.1$. The spectral density for the region below the threshold comprises of exact delta functions and is not plotted. Instead, the eigenvalues are shown as white lines. The red dashed lines represent the approximate "meson" spectrum from Bethe-Salpeter analysis in [5].

```
In[2]:= nullLevels[Dmax_]:=Module[
        {c=1/2,p=3,r=1,s=1,E0,series,nCut,level,factors},
        (* Shifting away the ground level *)
        E0=((p r - (p + 1) s)^2 - 1)/(4 p (p + 1)) - (c-1)/24;
        (* An estimate of the max n in the character q expansion. *)
        nCut=Max[
          Ceiling[(
            2 Sqrt[p (1 + p)] Sqrt[p^2 (1 + p)^2 (10 Dmax + E0)] +
            p (1 + p) (s + p (-r + s)))/(2 p^2 (1 + p)^2) // Abs],
          Ceiling[-((
            2 Sqrt[p (1 + p)] Sqrt[p^2 (1 + p)^2 (10 Dmax + E0)] +
            p (1 + p) (s + p (r + s)))/(2 p^2 (1 + p)^2)) // Abs]
        ];
        series=Sum[
          q^((2p (p+1) n +p r-(p+1) s)^2/(4p (p+1))-E0)
          - q^((2p (p+1) n +p r+(p+1) s)^2/(4p (p+1))-E0),
        {n,-nCut,nCut}];
        level=0;
        factors=1;
        Reap[While[level<=Dmax,
```

```
        Sow[level=Exponent[series-factors,q,List]//Min];
        factors=factors*(1-q^level);
    ]][[2,1]]
];
```

As an example, we compute the null levels up to the one next to 10

```
In[3]:= nullLevels[10]
```

```
Out[3]= {1,6,7,8,9,10,15}
```

and this means the vacuum character has factorization

$$\chi_{1,1} = \frac{q^{h_{1,1} - \frac{c-1}{24}}}{\eta(\tau)} (1 - q^1)(1 - q^6)(1 - q^7)(1 - q^8)(1 - q^9)(1 - q^{10})(1 - q^{15}) \cdots, \qquad (A.7)$$

suggesting the minimal basis should not include any of the above Virasoro generators.

As discussed above, we construct the minimal basis of monomials using the following code:

```
In[4]:= monomialsMinimal[deg_,Dmax_]:=monomialsMinimal[deg,Dmax]=Select[
        IntegerPartitions[deg],
        FreeQ[#,Alternatives@@nullLevels[Dmax]]&
    ]
```

We compute the Gram matrix restricted to the minimal basis of monomials. The states at different levels are orthogonal to each other, so the Gram matrix is block-diagonal in $\Delta$. For each pair of monomials, we take their inner product using the Virasoro algebra as coded in inner[]. The code that realizes this procedure is the following:

```
In[5]:= loadGramMatrixDmax[Dmax_]:=Module[
        {},
        gram=<||>;
        Do[
          gram[deg]=Table[
            If[j<=i,(* only compute the lower triangle *)
              inner[ Flatten[{
                Reverse[ monomialsMinimal[deg,Dmax][[i]] ],
                -monomialsMinimal[deg,Dmax][[j]]
              }] ],
              (*else*)0
            ],
            {i,Length[monomialsMinimal[deg,Dmax]]},
            {j,Length[monomialsMinimal[deg,Dmax]]}
          ];
          (* copy lower triangle to upper triangle *)
          gram[deg]=gram[deg]+Transpose[gram[deg]]
                    -DiagonalMatrix[Diagonal[gram[deg]]],
          {deg,2,Dmax}
        ]
      ]
```

Finally we construct the basis of quasi-primary states, which has 1-to-1 correspondence to the LCT basis (3). Each quasi-primary state is a "polynomial", i.e. a linear combination of monomial states, that is annihilated by the special conformal transformation generator $L_{+1}$. After we find these quasi-normal states, we orthonormalize these states using our Gram matrix, such that $\langle \mathcal{O}_l(0) | \mathcal{O}_{l'}(\infty) \rangle = 1$. The code that prepares the quasi-primary basis is the following

```
In[6]:= loadQuasiPrimaryDmax[Dmax_]:=Module[
        {LPlusOneMat,nsp},
        quasiPrimaries=<||>;
```

```
Do[
  LPlusOneMat=Table[
    inner[ Flatten[{
      Reverse[ monomialsMinimal[deg-1,Dmax][[i]] ],{+1},
      -monomialsMinimal[deg,Dmax][[j]]
    }] ],
    {i,Length[monomialsMinimal[deg-1,Dmax]]},
    {j,Length[monomialsMinimal[deg,Dmax]]}
  ];
  (* If there is no lower states for L_{+1} action to lower
  to, then every state in this level is quasiprimary *)
  If[Length[monomialsMinimal[deg-1,Dmax]]==0,
    nsp=IdentityMatrix[Length[monomialsMinimal[deg,Dmax]]],
    nsp=NullSpace[LPlusOneMat]
  ];
  quasiPrimaries[deg]=Orthogonalize[
    nsp,
    (* Too expensive to keep symbolic sqrts. Orthogonalize with
    finite but higher precision. *)
    #1 . N[gram[deg],40] . #2&
  ]
  ,
  {deg,2,Dmax}
  ]
]
```

Note that `loadQuasiPrimaryDmax[Dmax]` requires evaluating `loadGramMatrixDmax[Dmax]` first in order to have `gram` evaluated.

At the moment we do not have a proof that the factorization (A.5) works to all levels, nor do we have a proof that the minimal basis of monomials is linearly independent at all levels. In fact, the form (A.5) breaks down for other minimal models, suggesting that a general minimal basis has a more complex selection rule than that of Ising CFT.

## A.2 Matrix elements of $\sigma$

In this subsection we compute the matrix elements of $\int dx^- \sigma(x^-)\sigma(\infty)$. Given the form of the effective Hamiltonian and the fact that our basis states are Fourier transformation of CFT quasi-primary operators, these matrix elements are integrals of 4-point correlation functions. In practice, we compute the 4-point correlation functions of the type (37) and expand it into a Laurent series (38), reproduced here

$$\langle \mathcal{O}_l | \mathcal{O}_R(1)\mathcal{O}_R(y)|\mathcal{O}_{l'}\rangle = \sum_{m=-h_{l'}}^{h_l} a_m^{(ll')} y^m \,. \tag{A.8}$$

The integral formula (45) then maps each Laurent coefficient to a integral. Finally we sum up these integrals corresponding to a pair of quasi-primary state computed in the previous subsection, and weigh it by a proper normalization.

One is free to use any CFT techniques to obtain these 4-point correlation functions, but the complexity of the correlators scales up with $\Delta$ rapidly, which forces us to do some optimization in order to proceed. Because we constructed our basis using the Virasoro generators, we will continue to use Virasoro algebra to compute the correlators. We empasize that the Virasoro procedure is not the most efficient way to compute the matrix elements for IFT, and a much faster procedure based on the free fermion basis is introduced in Appendix B. Here our goal is

to present a simple and intuitive code that can quickly compute a reasonable number of matrix elements.

The Virasoro algebra code for the 4-point function is similar to that of the Gram matrix. We add a special structure `A[ops___]` to store the information of a commutator between a set of $L_n$'s and $\sigma(1)\sigma(y)$

$$
\begin{aligned}
\texttt{A[\_\_\_,n\_1,n\_2]}, &\quad \text{represents} \quad [L_{n_2}, [L_{n_1}, \cdots \sigma(1)\sigma(y) \cdots ]], \\
\texttt{A[]}, &\quad \text{represents} \quad \sigma(1)\sigma(y).
\end{aligned}
\tag{A.9}
$$

The Virasoro algebra that computes the contraction between `A[]` and the external Virasoro generators is coded as the following:

```
In[7]:= correlator[{_?Negative,z___}]=0;
        correlator[{x___,_?Positive}]=0;
        correlator[{}]=1;
        correlator[{x___,y1_,y2_,z___}]/;y1>y2:=correlator[{x,y1,y2,z}]=
        correlator[{x,y2,y1,z}]+If[
          y1+y2==0,
          (2y1*(-Plus[z]/.A[___]:>0)+cIsing/12 (y1^3-y1))correlator[{x,z}],
          (y1-y2)*correlator[{x,(y1+y2),z}]
        ];
        correlator[{front___,n_,A[ops___],back___}]/;n>0:=
          correlator[{front,A[ops,n],back}]+
          correlator[{front,A[ops],n,back}];
        correlator[{A[ops___],n_,back___}]/;n<0:=
          -correlator[{A[ops,n],back}]+correlator[{n,A[ops],back}];
        correlator[{A[ops___]}]:=evaluateCommutator[ops];
```

After all external $L_n$'s are contracted out, we send the commutator information to `evaluateCommutator[ops]` to evaluate it. The evaluator first use the Virasoro algebra to sort the $L_n$'s in the commutator into a canonical order

$$
\begin{aligned}
[\cdots [L_m, [L_n, \cdots]] \cdots] &= [\cdots [L_n, [L_m, \cdots]] \cdots] + [\cdots [[L_m, L_n], \cdots] \cdots] \\
&= [\cdots [L_n, [L_m, \cdots]] \cdots] + (m-n)[\cdots [L_{m+n}, \cdots] \cdots].
\end{aligned}
\tag{A.10}
$$

Note that the central piece of the $[L_m, L_n]$ commutator does not fire because commutators with a constant will vanish.

```
In[8]:= evaluateCommutator[front___,m_,n_,back___]/;m>n:=
        evaluateCommutator[front,m,n,back]=
          evaluateCommutator[front,n,m,back]-
          (m-n)evaluateCommutator[front,m+n,back];
        evaluateCommutator[ops___]/;OrderedQ[{ops}]:=
          applyCommutatorRule[ops];
```

When the commutator is in canonical order, we evaluate the commutators and obtain the function of $y$ using `applyCommutatorRule[ops]`. We work from the inside to the outside. At each depth we apply the rule

$$
\begin{aligned}
[L_n, \sigma(y_1)\sigma(y_2)] &= [L_n, \sigma(y_1)]\sigma(y_2) + \sigma(y_1)[L_n, \sigma(y_2)], \\
[\cdots [L_m, [L_n, \sigma(y_1)]\sigma(y_2)] \cdots] &= h_\sigma(n+1)y_1^n [\cdots [L_m\sigma(y_1)\sigma(y_2)] \cdots] \\
&\quad + y_1^{n+1} \frac{\partial}{\partial y_1}\Big( [\cdots [L_m\sigma(y_1)\sigma(y_2)] \cdots]\Big),
\end{aligned}
\tag{A.11}
$$

and similarly for $y_2$, with the identification $y_1 \to 1$ and $y_2 \to y$, until we obtain `applyCommutatorRule[]` representing $\langle \sigma(1)\sigma(y) \rangle = (1-y)^{-2h_\sigma}$ which is a common factor in all correlators. We omit this factor in the code.

```
In[9]:= hSig=1/16;
        applyCommutatorRule[m_,ops___]:=applyCommutatorRule[m,ops]=
          Collect[commutatorRule[Plus[ops],m][applyCommutatorRule[ops]],y];
        applyCommutatorRule[]=1;

        commutatorRule[NN_,n_][poly_]/;n>=0:=
          hSig(n+1)(1+y^n)poly+(-2hSig)Sum[y^k,{k,0,n}]poly+
          D[poly,y]y^(n+1)+Total[
            (NN-#)Coefficient[poly,y,#]y^#&/@
              Exponent[poly,y,List]
          ];
        commutatorRule[NN_,n_][poly_]/;n<=-1:=
          hSig(n+1)(1+y^n)poly+(2hSig)Sum[y^(-k-1),{k,0,-n-2}]poly+
          D[poly,y]y^(n+1)+Total[
            (NN-#)Coefficient[poly,y,#]y^#&/@
              Exponent[poly,y,List]
          ];
```

For example we can compute the correlator $\langle T|\sigma(1)\sigma(y)|T\rangle = 4\langle 0|L_2\sigma(1)\sigma(y)L_{-2}|0\rangle$:

```
In[10]:= 4 correlator[{2,A[],-2}] // Expand
```

```
Out[10]= 3/32 + 1/(64 y^2) + 7/(16 y) + (7 y)/16 + y^2/64
```

After we obtain the $y$-formula, we expand in Laurant series and apply the integral formula (45) to each power of $y$.

```
In[11]:= computeIntegralPow[m_,hl_,hlp_]:=
          computeIntegralPow[m,hl,hlp]=((Abs[m]+Abs[m+hlp-hl])/2);
        computeIntegral[poly_,hl_,hlp_]:=
          Coefficient[poly,y,#]*computeIntegralPow[#,hl,hlp]&/@
            Exponent[poly,y,List]//Total;
```

where $h_l = \Delta_l$ and $h_{l'} = \Delta_{l'}$ are the chiral conformal weights, similar to the scaling dimensions, of the in- and out- states, respectively. For the case of $\mathcal{O} = \mathcal{O}' = T$, the integral gives:

```
In[12]:= computeIntegral[4 correlator[{2,A[],-2}],2,2]
```

```
Out[12]= 15/16
```

The final wrapper to compute the full matrix of $\int dx^-\sigma(x^-)\sigma(\infty)$ is coded as the following. We first compute the matrix elements with respect to monomials, and dot the monomial matrix with the coefficients stored in quasiPrimaries to obtain matrix elements between a pair of quasi-primaries. The integral formula (45) also contains a $\Delta$-dependent weight, which also takes into account of the normalization of basis states (A.1). We apply this weight at the final step of constructing the quasi-primary matrix.

```
In[13]:= computeHeffSigma[Dmax_]:=Module[
          {statesL,statesR,primL,primR},
          quasiPrimariesNonZeroKeys=Select[
            quasiPrimaries,Length[#]>0&]//Keys;
          Module[{matElemMono,integrateCommutator},Table[
            statesL=monomialsMinimal[degL,Dmax];
            statesR=monomialsMinimal[degR,Dmax];
            primL=quasiPrimaries[degL];
            primR=quasiPrimaries[degR];
            (* Make a local version of correlator[] that does not store
            the correlator as y-series but do the Fourier transformation
            as soon as it can. This gives a significant speed-up. *)
            copyDefinitions[correlator,matElemMono];
```

```
    copyDefinitions[evaluateCommutator,integrateCommutator];
    (* Filter out memoised values *)
    DownValues[matElemMono]=DeleteCases[DownValues[matElemMono],
      HoldPattern[Verbatim[HoldPattern][
        matElemMono[{___Integer,A[___Integer],___Integer}]
      ] :> _]];
    DownValues[integrateCommutator]=
      DeleteCases[DownValues[integrateCommutator],
        HoldPattern[Verbatim[HoldPattern][
          integrateCommutator[___Integer]
        ] :> _]];
    (* Rewind the subroutines to avoid storing y-series *)
    matElemMono[{A[ops___]}]:=integrateCommutator[ops];
    integrateCommutator[ops___]/;OrderedQ[{ops}]:=
      computeIntegral[applyCommutatorRule[ops],degL,degR];
    (* Making the matrix *)
    primL . Table[
      matElemMono[{Reverse[stateL],A[],-stateR}//Flatten],
        {stateL,statesL},
        {stateR,statesR}
      ] . Transpose[primR] * factorDeltaWeight[degL,degR]

      ,
      {degL,Select[quasiPrimariesNonZeroKeys,#<=Dmax&]},
      {degR,Select[quasiPrimariesNonZeroKeys,#<=Dmax&]}
    ] ]//N (*Machine precision is enough*)
];

In[14]:= factorDeltaWeight[degL_,degR_]:=(2 Pi) ((-1)^(degL + degR) *
      Sqrt[Gamma[2 degL] Gamma[2 degR]])/Gamma[degL + degR - 1];
    copyDefinitions=ResourceFunction["CopyDefinitions"];
```

Note that the code requires evaluating loadQuasiPrimaryDmax[Dmax] first in order to have quasiPrimaries evaluated.

Now it is time to celebrate. We conclude this subsection by computing the spectrum (taking $g = 1$) using $P_+^{\text{eff}}$ at $\Delta_{\max} = 10$, which is a $7 \times 7$ matrix.

```
In[15]:= sigVev = 0.179886; (* Klassen-Melzer exact result, rounded *)
    loadGramMatrixDmax[10]; loadQuasiPrimaryDmax[10];
    Eigenvalues[
      sigVev * N[computeHeffSigma[10] // ArrayFlatten]
    ] // Sqrt // Sort

Out[15]= {1.65362, 2.68083, 3.37781, 4.28775, 4.74195, 7.11911, 7.45394}
```

The exact spectrum for the beginning 3 bound states are $(1.65288, 2.67441, 3.28765)$. We see that even at this low truncation, the spectrum is accurate to percentage level.

## A.3 The free fermion basis

We have computed the $\sigma$ deformation, and we also want to compute the $\epsilon$ deformation. It is both conceptually and computationally simple to consider $\epsilon$ as the fermion mass term and compute it with respect to the free fermion basis. Our strategy is to compute $\epsilon$ in the free fermion basis first and translate it to the Virasoro basis.

The free fermion basis is constructed from the fermion mode expansion in radial quantization

$$\psi(z) = \sum_{n \in \mathbb{Z}} a_{n+\frac{1}{2}} z^{-n-1}, \tag{A.12}$$

with the standard creation and annhilation operator

$$\left\{a_{n+\frac{1}{2}}, a_{m-\frac{1}{2}}\right\} = \delta_{n+m,0}. \tag{A.13}$$

Similar to the Virasoro basis, in the fermion basis there is a notion of "monomial states", which are the states created by a product of creation operators. We use the short-hand notation for an $n$-particle monomial

$$|\boldsymbol{k}\rangle \equiv |k_1, k_2, \cdots, k_n\rangle \equiv a_{-k_1-\frac{1}{2}} a_{-k_2-\frac{1}{2}} \cdots a_{-k_n-\frac{1}{2}} |0\rangle, \tag{A.14}$$

and we set the convention to be $k_1 > k_2 > \cdots > k_n \geq 0$. The basis is convenient as the inner product is diagonal

$$\langle \boldsymbol{k}|\boldsymbol{k}'\rangle = \prod_i \delta_{k_i, k_i'}. \tag{A.15}$$

A monomial state created by fermion radial quantization modes corresponds to acting a "monomial" of fermion local oprators on vacuum

$$|\partial^{\boldsymbol{k}}\psi(0)\rangle \equiv |\partial^{k_1}\psi\partial^{k_2}\psi\cdots\partial^{k_n}\psi(0)\rangle = \Gamma(\boldsymbol{k})|\boldsymbol{k}\rangle, \tag{A.16}$$

where $\Gamma(\boldsymbol{k}) \equiv \prod_i \Gamma(k_i)$, so we can swiftly switch between the radial quantization picture and the local operator picture. A state is characterized by the number of fermions $n$, and the "degree" $\ell \equiv \sum_i k_i \equiv |\boldsymbol{k}|$, which counts the total powers of derivatives. Since a free fermion has scaling dimension $\frac{1}{2}$, the scaling dimension of an operator is $\Delta = \frac{n}{2} + \ell$. The commutators between Virasoro generators and the creation operators are

$$\left[L_n, a_{m+\frac{1}{2}}\right] = -\left(\frac{1}{2}n + m + \frac{1}{2}\right) a_{m+n+\frac{1}{2}}. \tag{A.17}$$

We need to also find the orthonormal quasi-primary states in this basis. If we work with the fermion basis alone, the best algorithm will be the one provided in [4]; we do not use that algorithm here. Instead we will directly translate the Virasoro quasi-primary basis we constructed in the previous subsection to the fermion basis, in order to ensure that the $\sigma$ and $\epsilon$ deformations are evaluated on the same basis.

We begin by defining a basis of free fermion monomials at level $\Delta$. The basis consists all non-repetitive integer partition of $\Delta - n/2$ into $n$ numbers. We only need the bosonic states with $n$ even, as the fermionic sector decouple.

```
In[16]:= monomialsFermionAll[Delta_] :=
    monomialsFermionAll[Delta] = Join @@ Table[
        Select[IntegerPartitions[Delta + n/2, {n}] - 1,
          DuplicateFreeQ],
        {n, 2, 1/2*(1 + Sqrt[1 + 8 Delta]), 2}
    ]
```

We can use the commutator (A.17) to compute the inner product between a Virasoro monomial and a fermion monomial $\langle L_{+S}|\boldsymbol{k}\rangle$. The code is similar to that of the pure Virasoro case.

```
In[17]:= innerLPsi[{}, {x_, ___}] /; x <= 0 := 0;
    innerLPsi[{}, {}] := 1;
    innerLPsi[{a___}, {x___, y1_, y2_, z___}] /; y1 > y2 :=
        (-1) innerLPsi[{a}, {x, y2, y1, z}] +
        KroneckerDelta[y1 + y2, 1] innerLPsi[{a}, {x, z}];
    innerLPsi[{a___}, {x___, y1_, y2_, z___}] /; y1 == y2 := 0;
    innerLPsi[{a___}, {x___, y1_}] /; y1 > 0 := 0;
    innerLPsi[{a___, b_}, xl_List] := innerLPsi[{a, b}, xl] = Sum[
        With[{k = xl[[i]]},
```

```
        (-1/2 b - k + 1/2) *
          innerLPsi[{a}, ReplacePart[xl, i -> b + k]]
      ], {i, Length[xl]}
    ]
```

In Appendix A.1 we have computed the quasi-primary basis in terms of the Virasoro monomials $|\mathcal{O}_l\rangle = \sum_S c_S |L_{-S}\rangle$. We multiply those to the inner product matrix $\langle L_{+S}|\boldsymbol{k}\rangle$ to obtain the same quasi-primary states in terms of the fermion monomials $\langle \mathcal{O}_l|\boldsymbol{k}\rangle$.

```
In[18]:= loadQuasiPrimaryFermions[Dmax_]:=Module[
        {LPhiInnerMat},
          quasiPrimariesInFermions=<||>;
          Do[
            LPhiInnerMat = Table[
              innerLPsi[Reverse[stateL],-statePhi],
              {stateL,monomialsMinimal[Delta,Dmax]},
              {statePhi,monomialsFermionAll[Delta]}
            ];
            quasiPrimariesInFermions[Delta]=
              quasiPrimaries[Delta] . LPhiInnerMat;,
            {Delta,DeleteCases[quasiPrimaries,{}]//Keys}
          ]
        ]
```

The code requires first running loadQuasiPrimaryDmax[Dmax] before evaluation.

## A.4 Matrix elements of $\epsilon$

In this subsection we compute the fermion mass term. The matrix element between a pair of monomials is the Fourier transformation of the 3-point function (the superscript "−" is omitted and we take units where $p = 1$)

$$M_{\boldsymbol{k},\boldsymbol{k}'} \equiv \int dx\,dy\,dz\, e^{ip(x-z)} \partial^{\boldsymbol{k}} \psi(x) : \psi \frac{1}{i\partial} \psi : (y) \partial^{\boldsymbol{k}'} \psi(z). \tag{A.18}$$

The building block for such a spacial 3-point function is the Wick contraction between a pair of fermions

$$\langle \partial^k \psi(x) \partial^{k'} \psi(z) \rangle = \frac{(-1)^k \Gamma(k+k'+1)}{(x-z)^{k+k'+1}}. \tag{A.19}$$

The total result of the 3-point function is a sum of all possible Wick contractions times the fermion permutation signs

$$\partial^{\boldsymbol{k}} \psi(x) : \psi \frac{1}{i\partial} \psi : (y) \partial^{\boldsymbol{k}'} \psi(z) = \sum_{i,j} (-1)^{i-j} \frac{(-1)^{k_i} \Gamma(k_i+1)}{(x-y)^{k_i+1}} \frac{1}{i\partial} \frac{\Gamma(k_i+1)}{(y-z)^{k'_j+1}} \frac{A_{\boldsymbol{k}/k_i,\boldsymbol{k}'/k'_j}}{(x-z)^{\Delta+\Delta'-k_i-k'_j-1}} \tag{A.20}$$
$$+ (k_i \leftrightarrow k'_j),$$

where $\boldsymbol{k}/k_i$ means the vector resulting from removing $k_i$ from $\boldsymbol{k}$ and $\Delta$ and $\Delta'$ are the scaling dimensions of the monomials at $x$ and $z$, respectively. In the above formula, we select one particle $\partial^{k_i}\psi$ from the $\partial^{\boldsymbol{k}}\psi(x)$ operator, and one particle $\partial^{k'_j}\psi$ from the other side to contract with the operator at $y$. We call this the "active part" of the 3-point functions. The rest of the particles at $x$ are $\partial^{\boldsymbol{k}/k_i}\psi$, who contract with the other side $\partial^{\boldsymbol{k}'/k'_j}\psi$ in all possible ways, giving what we call the "spectator part". The numerical coefficient of the spectator part can be evaluated recursively

$$A_{\boldsymbol{k},\boldsymbol{k}'} = \sum_j (-1)^{k_1} (-1)^{n-j} \Gamma(k_1 + k'_j + 1) A_{\boldsymbol{k}/k_1,\boldsymbol{k}'/k'_j}. \tag{A.21}$$

The recursion of $A_{k,k'}$ is realized by the following code. The signs from taking spacial derivatives are omitted because they cancel out in the final result.

```
In[19]:= wickContract[{}, {}] = 1;
         wickContract[k1_List, k2_List] :=
         wickContract[k1, k2] = Block[{n = Length[k2], num}, Sum[
           (-1)^(n - i)*
           Gamma[k1[[1]] + k2[[i]] + 1] *
             wickContract[Drop[k1, {1}], Drop[k2, {i}]],
           {i, Range[n] }]
         ];
```

The Fourier transformation (A.20) has an IR singularity. We defer the detailed discussion of the nature of the singularity and its regularization scheme to Appendix D and here we just state that the final result is the following

$$M_{k,k'} = \sum_{i,j}(-1)^{i-j}\frac{A_{k/k_i, k'/k'_j}}{\Gamma(\Delta + \Delta' - 1)} \times \begin{cases} \Gamma(k_i + k'_j), & k_i + k'_j > 0, \\ -H_{\Delta+\Delta'-2} - \log\epsilon, & k_i + k'_j = 0, \end{cases} \quad (A.22)$$

where $H_k = \sum_{n=1}^{k} \frac{1}{n}$ is the Harmonic number, and $\log\epsilon$ is the IR regulator. We will need to change the value of $\log\epsilon$, so we will code up two separate subroutines that compute the finite part and the coefficient of $\log\epsilon$ of $M_{k,k'}$.

```
In[20]:= massTermRegular[k_, kp_]/;Length[k]!=Length[kp] := 0;
         massTermRegular[k_, kp_] := massTermRegular[k, kp]=
         With[{n = Length[k]}, Sum[
           If[k[[i]]==0&&kp[[j]]==0,0,
             (-1)^(i-j) * (-Signature[Range[n]//Reverse]) *
             Gamma[k[[i]] + kp[[j]]]*
             wickContract[Drop[k, {i}], Drop[kp, {j}]]
           ],
           {i, Range[n]}, {j, Range[n]}
         ] ]
         massTermSingular[k_, kp_]/;Length[k]!=Length[kp] := 0;
         massTermSingular[k_, kp_] := massTermSingular[k, kp] =
         With[{n = Length[k]}, Sum[
           If[k[[i]]==0&&kp[[j]]==0,
             (-1)^(i-j) * (-Signature[Range[n]//Reverse]) *
             wickContract[Drop[k, {i}], Drop[kp, {j}]],
             0
           ],
           {i, Range[n]}, {j, Range[n]}
         ] ]
```

Finally we define wrapper functions that compute $M_{k,k'}$ for all states at all levels, dot them with the quasi-primary coeifficients and weigh by proper normalization factors.

```
In[21]:= computeMassMatrixFinite[Dmax_]:=Module[
           {statesL,statesR,primL,primR},
           quasiPrimariesNonZeroKeys=Select[
             quasiPrimaries,Length[#]>0&]//Keys;
           Table[
             statesL=monomialsFermionAll[DeltaL];
             statesR=monomialsFermionAll[DeltaR];
             primL=#/factorsFermion[DeltaL]&/@
                   quasiPrimariesInFermions[DeltaL];
             primR=#/factorsFermion[DeltaR]&/@
                   quasiPrimariesInFermions[DeltaR];
```

```
        primL . Table[
          massTermRegular[stateL,stateR]
          - massTermSingular[stateL, stateR] *
            HarmonicNumber[DeltaL+DeltaR-2],
            {stateL,statesL},
            {stateR,statesR}
          ] . Transpose[primR] *
          factorDeltaWeightMass[DeltaL,DeltaR],
        {DeltaL,Select[quasiPrimariesNonZeroKeys,#<=Dmax&]},
        {DeltaR,Select[quasiPrimariesNonZeroKeys,#<=Dmax&]}
      ]
    ];

In[22]:= computeMassMatrixLogDivergence[Dmax_]:=Module[
        {statesL,statesR,primL,primR},
        quasiPrimariesNonZeroKeys=Select[
          quasiPrimaries,Length[#]>0&]//Keys;
        Table[
          statesL=monomialsFermionAll[DeltaL];
          statesR=monomialsFermionAll[DeltaR];
          primL=#/factorsFermion[DeltaL]&/@
                quasiPrimariesInFermions[DeltaL];
          primR=#/factorsFermion[DeltaR]&/@
                quasiPrimariesInFermions[DeltaR];
          primL . Table[
            massTermSingular[stateL, stateR],
              {stateL,statesL},
              {stateR,statesR}
            ] . Transpose[primR] *
            factorDeltaWeightMass[DeltaL,DeltaR],
          {DeltaL,Select[quasiPrimariesNonZeroKeys,#<=Dmax&]},
          {DeltaR,Select[quasiPrimariesNonZeroKeys,#<=Dmax&]}
        ]//N
      ];

In[23]:= factorDeltaWeightMass[DeltaL_,DeltaR_]:=
        Sqrt[Gamma[2DeltaL] Gamma[2DeltaR]]/Gamma[DeltaL+DeltaR-1];
        factorsFermion[Delta_]:=factorsFermion[Delta]=
          (Times@@Gamma[#+1])&/@monomialsFermionAll[Delta]
```

The normalization come from (A.1), (A.16), and the denominator of (A.22).

## A.5 Running the code at finite $\eta$

In this subsection we show how to compute the LCT effective Hamiltonian for generic $\eta$ in practice. We take $\langle\sigma\rangle$ from the result in [11] and plug in $\eta = 2$. We work in the unit that $g = 1$.

```
In[24]:= eta = 2;
        sigmaVevEta2 = 1.504119;
```

Then we use the code in the previous subsections to compute the basis states and matrix elements. Note that for the mass term we have two separate contributions: the finite part and the logarithmic IR divergent part $\propto \log \epsilon$. It takes about 30s for the code to compute everything for $\Delta_{\max} = 14$ on a laptop.

```
In[25]:= loadGramMatrixDmax[14]
        loadQuasiPrimaryDmax[14]
```

```
loadQuasiPrimaryFermions[14]
sigmaMat = computeHeffSigma[14] // ArrayFlatten;
massMatFinite = computeMassMatrixFinite[14] // ArrayFlatten;
massMatLogDivergence =
    computeMassMatrixLogDivergence[14] // ArrayFlatten;
```

We need to find the value for the IR regulator $\epsilon$, which is the cutoff of the fermion parton momentum. According to the discussion in Appendix D, there exists a Goldilocks IR regulator value which depends on $\Delta_{\max}$ and $\eta$. In the infinite truncation limit $\Delta_{\max} \to \infty$ the IR regulator also goes to infinity. At finite $\Delta_{\max}$ the values are selected such that the convergence of the spectrum is the fastest. We take the ansatz that

$$\log \epsilon = -2 \log \Delta_{\max} + \xi(m). \tag{A.23}$$

The criterion of "fastest convergence" is the following: At the Goldilocks IR cutoff $\xi_*$, if we diagonalize $P_+^{\text{eff}}$ at our largest truncation $\Delta_{\max}$ and a slightly lower truncation $\Delta_{\max} \to \Delta_{\max}-2$, the lowest eigenvalue from the two truncations are the same. In practice, the dependence of eigenvalues on $\xi$ is unknown but smooth, and we can use Newton's method to find $\xi_*$. The Newton's method code is the following:

```
In[26]:= findZero[fn_,{x0_,y0_},{x1_,y1_},eps_]:=Module[
            {xNew=(x1 y0-x0 y1)/(y0-y1),yNew},
            yNew=fn[xNew];
            If[Abs[yNew]<eps,
                {xNew,yNew},
                findZero[fn,{xNew,yNew},{x0,y0},eps] ]
        ];
        findZero[fn_,x0_,x1_,eps_]:=findZero[fn,{x0,fn[x0]},{x1,fn[x1]},eps]
```

We use the Newton's method to search for the $\xi_*$ that satisfies the Goldilocks criterion.

```
In[27]:= IRCutValEta2=Module[
            {hamIRCut,diff,idx},
            idx[dmax_]:=Range[1,Total[Length/@KeySelect[quasiPrimaries,#<=dmax&]]];
            hamIRCut[dmax_,xi_]:=(sigmaVevEta2*sigmaMat+
                eta^2(massMatFinite+
                    (2Log[dmax]+xi)massMatLogDivergence))[[idx[dmax],idx[dmax]]];
            diff[xi_]:=Min[Eigenvalues[hamIRCut[12,xi]]]
                -Min[Eigenvalues[hamIRCut[14,xi]]];
            findZero[diff,4,9,10^-5][[1]]
        ]

Out[27]= 5.08052
```

We find a numerical value for $\xi_*$. For other values of $\eta$, we will need to run Newton's solver again to find the Goldilocks IR cutoff value. A solution is not always guaranteed because our ansatz is an approximation that works only when $\Delta_{\max}$ is sufficiently large. If this happens, one can take higher truncation and try again or stay in the same truncation and use $\xi_*$ extrapolated from other values of $\eta$. The result is not sensitive to small changes in the IR cutoff.

Now we can substitute it in $P_+^{\text{eff}}$ and diagonalize to obtain the bound state spectrum:

```
In[28]:= Eigenvalues[
            sigmaVevEta2 * sigmaMat + eta^2 * (massMatFinite+
                (2Log[14]+IRCutValEta2)massMatLogDivergence)
        ]//Sqrt//TakeSmallest[4]

Out[28]= {7.51354, 9.92873, 11.785, 13.5017}
```

These values match the results in [5, 11] to percentage precision, despite the low truncation $\Delta_{\max} = 14$.

# B  A fast algorithm for computing matrix elements

In order to use the effective Hamiltonian interaction term $\sigma(x)\sigma(\infty)$ in practice, we need to be able to compute its matrix elements between basis states. Because our basis states are all operators built out of the chiral algebra (they are built from products and derivatives of the stress tensor $T$), one approach is simply to use the chiral algebra to compute any correlator of the form

$$\langle \mathcal{O}_l(x)\sigma(y_1)\sigma(y_2)\mathcal{O}_{l'}(z)\rangle, \tag{B.1}$$

where $\mathcal{O}_l, \mathcal{O}_{l'}$ are the basis state operators. Without loss of generality, one can set $x = \infty$ and $z = 0$, compute the correlator by using the commutation relations of the algebra with $\sigma(y)$, and then restore $x$ and $z$ using conformal transformations. Simply doing a brute force calculation along these lines is a practical method up until roughly $\Delta_{\max} \sim 25$. However, one can do the computation more efficiently by taking advantage of the free fermion representation of the basis operators.

The standard expansion of NS sector modes of a chiral fermion field is

$$\psi(z) = \sum_{n\in\mathbb{Z}} a_{n+\frac{1}{2}} z^{-n-1}. \tag{B.2}$$

The $a_{n+\frac{1}{2}}$ operators are all creation operators on the circle for $n < 0$ and annihilation operators for $n > 0$. They are therefore convenient for constructing the physical space of primary states because they have simple commutation relations with the generators of the conformal algebra, but they do not have simple commutation relations with $\sigma(y)$. Since we are interested in computing correlators of the form $\langle \mathcal{O}_l|\sigma(y_1)\sigma(y_2)|\mathcal{O}_{l'}\rangle$, we will also make use of the following mode decomposition of $\psi(z)$:

$$\psi(z) = \sum_{\ell\in\mathbb{Z}} c_\ell \frac{z^{-\ell}}{\sqrt{(y_1-z)(y_2-z)}}. \tag{B.3}$$

The advantage of this basis is that the operators $\sigma(y_1)$ and $\sigma(y_2)$ become transparent to the $c_\ell$ operators, i.e., $[c_\ell, \sigma(y_1)\sigma(y_2)] = 0$, because the square roots in the denominator remove the branch cut singularities due to the $\sigma$ operators. One can see this explicitly from, say, the correlator

$$\langle \psi(x)\sigma(y_1)\sigma(y_2)\psi(z)\rangle = -\frac{1}{|y_1-y_2|^{\frac{1}{4}}(x-z)}\left(\frac{\sqrt{\frac{(x-y_1)(y_2-z)}{(y_1-z)(x-y_2)}} + \sqrt{\frac{(y_1-z)(x-y_2)}{(x-y_1)(y_2-z)}}}{2}\right). \tag{B.4}$$

The idea of the more efficient algorithm is to first construct the primary basis states $|\mathcal{O}_l\rangle$ in terms of the $a_{n+\frac{1}{2}}$ operators, and then to decompose the $a_{n+\frac{1}{2}}$ in terms of the $c_\ell$ operators. Once the primary basis states are written in terms of the $c_\ell$ operators, it is straightforward to compute the correlators $\langle \mathcal{O}_l|\sigma(y_1)\sigma(y_2)|\mathcal{O}_{l'}\rangle$ because the $c_\ell$'s commute with the $\sigma$ operators. Finally we compute the LCT matrix elements using the formula (45).

To complete the prescription, we need to find decompositions of $a_{n+\frac{1}{2}}$s into $c_\ell$s, as well as the anti-commutation relations of the $c_\ell$s. The anti-commutation relations of the $a_{n+\frac{1}{2}}$s are just the standard ones:

$$\left\{a_{n+\frac{1}{2}}, a_{m-\frac{1}{2}}\right\} = \delta_{n+m,0}. \tag{B.5}$$

We can obtain a decomposition of $a_{n+\frac{1}{2}}$s into $c_\ell$s, and vice versa, directly from comparing the Taylor series of (B.2) and (B.3) around $z = 0$,

$$a_{n-\frac{1}{2}} = \sum_{m=0}^{\infty} c_{n+m} f_m(y_1, y_2), \quad c_\ell = \sum_{m=0}^{\infty} a_{\ell+m-\frac{1}{2}} g_m(y_1, y_2), \tag{B.6}$$

and another decomposition by expanding around $z = \infty$:

$$a_{n+\frac{1}{2}} = \sum_{m=0}^{\infty} c_{n-m} \widetilde{f}_m(y_1, y_2), \quad c_\ell = \sum_{m=0}^{\infty} a_{\ell-m+\frac{1}{2}} \widetilde{g}_m(y_1, y_2), \tag{B.7}$$

where the coefficients are

$$f_m(y_1, y_2) = \frac{(\frac{1}{2})_m}{m!} y_1^{-\frac{1}{2}} y_2^{-\frac{1}{2}-m} {}_2F_1\left(\frac{1}{2}, -m; \frac{1}{2} - m, \frac{y_2}{y_1}\right), \tag{B.8}$$

$$g_m(y_1, y_2) = \frac{(-\frac{1}{2})_m}{m!} y_1^{\frac{1}{2}} y_2^{\frac{1}{2}-m} {}_2F_1\left(-\frac{1}{2}, -m; \frac{3}{2} - m, \frac{y_2}{y_1}\right),$$

$$\widetilde{f}_m(y_1, y_2) = \frac{(\frac{1}{2})_m}{m!} y_1^m {}_2F_1\left(\frac{1}{2}, -m; \frac{1}{2} - m, \frac{y_2}{y_1}\right), \tag{B.9}$$

$$\widetilde{g}_m(y_1, y_2) = \frac{(-\frac{1}{2})_m}{m!} y_1^m {}_2F_1\left(-\frac{1}{2}, -m; \frac{3}{2} - m, \frac{y_2}{y_1}\right).$$

By inspection of (B.3), we see that the $c_\ell$ modes are singular (regular) at $z = 0$ if $\ell > 0$ ($\ell < 0$) and at $z = \infty$ if $\ell < 0$ ($\ell > 0$). Therefore,

$$c_\ell |0\rangle = 0 \qquad (\ell > 0), \qquad \langle 0| c_\ell = 0 \qquad (\ell < 0). \tag{B.10}$$

Now we derive the anti-commutators $\{c_n, c_m\}$. First, we substitute (B.6) into the anti-commutator

$$\{c_n, c_m\} = \sum_{p=0}^{\infty} \sum_{q=0}^{\infty} g_p(y_1, y_2) g_q(y_1, y_2) \{a_{n+p-1/2}, a_{m+q-1/2}\}. \tag{B.11}$$

Because $\{a_{n+p-1/2}, a_{m+q-1/2}\} = \delta_{m+p+n+q-1, 0}$, and $p, q \geq 0$, the sum is nonzero only if $m + n \leq 1$. In particular,

$$\{c_n, c_m\} = \begin{cases} 0, & m + n > 1, \\ g_0(y_1, y_2) g_0(y_1, y_2) = y_1 y_2, & m + n = 1, \\ 2g_0(y_1, y_2) g_1(y_1, y_2) = -(y_1 + y_2), & m + n = 0, \\ \cdots \end{cases} \tag{B.12}$$

On the other hand, we get another relation by substituting the decomposition (B.7) from large $z$ into the anti-commutator:

$$\{c_n, c_m\} = \sum_{p=0}^{\infty} \sum_{q=0}^{\infty} \widetilde{g}_p(y_1, y_2) \widetilde{g}_q(y_1, y_2) \{a_{n-p+1/2}, a_{m-q+1/2}\}. \tag{B.13}$$

Because $\{a_{n-p+1/2}, a_{m-q+1/2}\} = \delta_{m-p+n-q+1, 0}$, and $p, q \geq 0$, the sum is nonzero only if $m + n \geq -1$. Combining with (B.12), we have

$$\{c_n, c_m\} = \begin{cases} 0, & m + n > 1, \\ g_0(y_1, y_2) g_0(y_1, y_2) = y_1 y_2, & m + n = 1, \\ 2g_0(y_1, y_2) g_1(y_1, y_2) = -(y_1 + y_2), & m + n = 0, \\ \widetilde{g}_0(y_1, y_2) \widetilde{g}_0(y_1, y_2) = 1, & m + n = -1, \\ 0, & m + n < -1 \end{cases} \tag{B.14}$$

$$= y_1 y_2 \delta_{m+n-1, 0} - (y_1 + y_2) \delta_{m+n, 0} + \delta_{m+n+1, 0}.$$

As a special case,

$$c_0^2 = -\frac{1}{2}(y_1 + y_2).\tag{B.15}$$

The correlator $\langle \mathcal{O}_l | \sigma(y_1)\sigma(y_2) | \mathcal{O}_{l'} \rangle$ is homogeneous and symmmetric between $y_1$ and $y_2$, and it is always in the form of a polynomial divided by some power of $y_1 y_2$. It is easy to check that the degree of the function is $\Delta_l - \Delta_{l'}$.

# C Expansion of effective action up to $O(g^4)$

In this appendix, we will perform some explicit checks of our formalism in a perturbative expansion in the deforming operator up to $O(g^4)$.

## C.1 Dyson series vs Schur complement $(P_+)_{\text{eff}}$

Here, we will see explicitly in perturbation theory how the formulation of $(P_+)_{\text{eff}}$ in terms of the Dyson series matches the formulation in terms of the Schur complement. We take the original Hamiltonian $P_+$ in the full theory (including non-chiral modes) to be the Hamiltonian of the UV CFT plus the relevant deformations, minus a constant term to fix the vacuum energy of the deformed theory to zero:

$$P_+ = (P_+)_{\text{CFT}} + gV(0) - \mathbb{1}E_0(g), \quad V(t) \equiv \int dx\, \mathcal{O}_R(t,x).\tag{C.1}$$

We can always subtract out the vacuum energy $E_0$ of the deformed theory without loss of generality. In an expansion in $g$, for simplicity we will assume that only even powers appear due to a $\mathcal{O}_R \to -\mathcal{O}_R$ symmetry (as with the $\sigma$ deformation of Ising):

$$E_0 = g^2 E_0^{(2)} + g^4 E_0^{(4)} + \dots\tag{C.2}$$

The Dyson series formulation (2) has an expansion of the form

$$(P_+)_{\text{eff}} = \lim_{t \to \infty(1-i\epsilon)} \frac{1}{U(t,0)} \langle l | (g\mathcal{O}_R(t) - E_0(g))(1 - i\int_0^t dt_1(g\mathcal{O}_R(t_1) - E_0(g)) + \dots) | l' \rangle.\tag{C.3}$$

On the other hand, the large momentum Schur complement formulation from [2] is purely algebraic:

$$(P_+)_{\text{eff}} = Z^{-\frac{1}{2}}\left((P_+)_{ll} - g^2 V_{lh}\frac{1}{(P_+)_{hh}}V_{hl}\right)Z^{-\frac{1}{2}},\tag{C.4}$$

where $P_+ = \begin{pmatrix} (P_+)_{ll} & (P_+)_{lh} \\ (P_+)_{hl} & (P_+)_{hh} \end{pmatrix}$ breaks $P_+ \equiv \frac{H - P_x}{\sqrt{2}}$ into the 'light' sector of chiral descendants of the vacuum and the 'heavy' sector of all other states, and

$$Z \equiv \mathbb{1} + (P_+)_{lh}\frac{1}{(P_+)_{hh}^2}(P_+)_{hl}.\tag{C.5}$$

At $O(g^2)$, it is not too much work to see that the Dyson series formulation for $(P_+)_{\text{eff}}$ matches its Schur complement formulation. Simply expand out the Dyson series to $O(g^2)$, insert a complete space of states between any insertions of the interaction $V(t)$, and then

perform the integrations over time, as follows:

$$
\begin{aligned}
(P_+)_{\text{eff}}|_{g^2} &= g^2 \lim_{t \to \infty(1-i\epsilon)} \left[ -E_0^{(2)} \delta_{ll'} - i \langle l|V(t) \int_0^t dt_1 V(t_1)|l'\rangle \right] \\
&= g^2 \lim_{t \to \infty(1-i\epsilon)} \left[ -E_0^{(2)} \delta_{ll'} - i \sum_n \int_0^t dt_1 \langle l|V(0)|n\rangle e^{iE_n(t_1-t)} \langle n|V(0)|l'\rangle \right] \\
&= g^2 \left[ -E_0^{(2)} \delta_{ll'} - \sum_n \frac{\langle l|V(0)|n\rangle \langle n|V(0)|l'\rangle}{E_n} \right].
\end{aligned}
\tag{C.6}
$$

This exactly matches the Schur complement formula expanded out to $O(g^2)$:

$$
(P_+)_{\text{eff}}|_{g^2} = g^2 \left( -E_0^{(2)} \delta_{ll'} - V_{lh} \frac{1}{(P_+)_{hh}} V_{hl'} \right),
\tag{C.7}
$$

because $\langle l|V(0)|n\rangle$ vanishes if $\langle l|$ and $|n\rangle$ are chiral vacuum descendants, so only the 'heavy' sector contributes to the sum on $n$, and the CFT eigenvalues $E_n$ for heavy states are the eigenvalues of $P_{hh}$ at $O(g^0)$.

Explicitly checking that the two formulations are equal becomes more tedious as we proceed to higher orders. We will perform this explicit check at $O(g^4)$. We need to expand the factors $Z^{-\frac{1}{2}}$ in the Schur complement out to $O(g^2)$, and the central terms in parenthesis out to $O(g^4)$:

$$
\begin{aligned}
(P_+)_{\text{eff}} =\ & \left( \delta_{ll''} - \frac{1}{2} g^2 V_{ln} \frac{1}{E_n^2} V_{nl''} \right) \\
& \times \left( -\delta_{l''l'''} E_0(g) - g^2 V_{l''n} \left( \left( \frac{1}{E_n} + \frac{E_0(g)}{E_n^2} \right) \delta_{nn''} + \frac{1}{E_n} g V_{nn'} \frac{1}{E_{n'}} g V_{n'n''} \frac{1}{E_{n''}} \right) V_{n''l'''} \right) \\
& \times \left( \delta_{l'''l''} - \frac{1}{2} g^2 V_{l'''n} \frac{1}{E_n^2} V_{nl'} \right) + O(g^6).
\end{aligned}
\tag{C.8}
$$

Repeated indices are summed over in the above equation. The $O(g^4)$ piece of this is

$$
\begin{aligned}
(P_+)_{\text{eff}}|_{g^4} &= g^4 \left( E_0^{(4)} \delta_{ll'} - V_{ln} \frac{1}{E_n} V_{nn'} \frac{1}{E_{n'}} V_{n'n''} \frac{1}{E_{n''}} V_{n''l'} \right. \\
&\qquad \left. + \frac{1}{2} V_{ln} \frac{1}{E_n^2} V_{nl''} V_{l''n'} \frac{1}{E_{n'}} V_{n'l'} + \frac{1}{2} V_{ln'} \frac{1}{E_{n'}} V_{n'l''} V_{l''n} \frac{1}{E_n^2} V_{nl'} \right) \\
&= g^4 \left( -E_0^{(4)} \delta_{ll'} - \frac{V_{ln} V_{nn'} V_{nn''} V_{n''l'}}{E_n E_{n'} E_{n''}} + \frac{V_{ln} V_{nl''}}{E_n^2} \frac{V_{l''n''} V_{n''l'}}{E_{n''}} \right).
\end{aligned}
\tag{C.9}
$$

On the other hand, if we expand out the Dyson series to $O(g^4)$, we find

$$
\begin{aligned}
(P_+)_{\text{eff}}|_{g^4} =\ & g^4 \left( (-i)^3 \langle l|V(t) \int_0^t dt_1 \int_0^{t_1} dt_2 \int_0^{t_2} dt_3 V(t_1)V(t_2)V(t_3)|l'\rangle - E_0^{(4)} \delta_{ll'} \right. \\
& + E_0^{(2)} \langle l| \int_0^t dt_1 \int_0^{t_1} dt_2 V(t_1)V(t_2)|l'\rangle \\
& - i E_0^{(2)} E_0^{(2)} t + E_0^{(2)} t \langle l|V(t) \int_0^t dt_1 V(t_1)|l'\rangle \\
& \left. - \langle l| \left( -iV(t) \int_0^t dt_1 V(t_1) - E_0^{(2)} \right) \left( it E_0^{(2)} - \int_0^t dt_1 \int_0^{t_1} dt_2 V(t_1)V(t_2) \right) |l'\rangle \right).
\end{aligned}
\tag{C.10}
$$

We have used the expansion of $U(t)$:

$$
\begin{aligned}
U(t) &= 1 - i\int_0^t dt(gV(t) - E_0(g)) - \int_0^t dt_1 \int_0^{t_1} dt_2(gV(t_1) - E_0(g))(gV(t_2) - E_0(g)) \\
&= 1 + ig^2 t E_0^{(2)} - g^2 \int_0^t dt_1 \int_0^{t_1} dt_2 V(t_1)V(t_2),
\end{aligned}
\tag{C.11}
$$

as well as the fact that the energy of the light modes is zero in the undeformed theory. Now, note that the terms linear in $t$ manifestly cancel:

$$
\begin{aligned}
(P_+)_{\text{eff}}|_{g^4} = g^4\Bigg( &i\langle l|V(t)\int_0^t dt_1 \int_0^{t_1} dt_2 \int_0^{t_2} dt_3 V(t_1)V(t_2)V(t_3)|l'\rangle - E_0^{(4)}\delta_{ll'} \\
&- i\langle l|(V(t)\int_0^t dt_1 V(t_1))|l''\rangle\langle l''|(\int_0^t dt_2 \int_0^{t_2} dt_3 V(t_2)V(t_3))|l'\rangle\Bigg).
\end{aligned}
\tag{C.12}
$$

Finally, the last step is to insert a complete set of states and do the time integrals. We take $t$ to be in a range where $t \gg 1/E$ if $E$ is the CFT $P_+$ energy of a heavy mode and $t \ll 1/E$ if $E$ is the CFT $P_+$ energy of a light mode. In practice, this means that an integral of the form

$$
\int_0^{t_i} dt\, e^{iE_n t} = \frac{1}{iH}(e^{iE_n t_i} - 1) = \left\{ \begin{array}{ll} \frac{i}{E_n} & n \text{ heavy} \\ t_i & n \text{ light} \end{array} \right\}.
\tag{C.13}
$$

In the above expression, if we insert a complete set of states between every insertion of $V$, light states only contribute to insertions of the form $\sim \langle l|VV|n\rangle\langle n|VV|l'\rangle$ since the vacuum is not connected to light states by an odd number of insertions of $V$ due to the assumed $V \to -V$ symmetry. The integrals in the first line of (C.12) become

$$
\begin{aligned}
&i\langle l|V(t)\int_0^t dt_1 \int_0^{t_1} dt_2 \int_0^{t_2} dt_3 V(t_1)V(t_2)V(t_3)|l'\rangle \\
&= -\frac{V_{ln}V_{nn'}V_{nn''}V_{n''l'}}{E_n E_{n'} E_{n''}} + (V_{ln}V_{nl''}V_{l''n''}V_{n''l'})\left(\frac{1}{E_n E_{n''}^3} + \frac{1}{E_n^2 E_{n''}} - \frac{it}{E_n E_{n''}}\right),
\end{aligned}
\tag{C.14}
$$

where the sum on $n$ is only over heavy states and the sum on $l$ is only over light states. The integrals in the second line of (C.12) become

$$
\begin{aligned}
&-i\langle l|(V(t)\int_0^t dt_1 V(t_1))|l''\rangle\langle l''|\left(\int_0^t dt_2 \int_0^{t_2} dt_3 V(t_2)V(t_3)\right)|l'\rangle \\
&= (V_{ln}V_{nl''}V_{l''n''}V_{n''l'})\left(-\frac{1}{E_n E_{n''}^3} + \frac{it}{E_n E_{n''}}\right).
\end{aligned}
\tag{C.15}
$$

Combining all terms, we see that the Dyson series at fourth order is

$$
(P_+)_{\text{eff}}|_{g^4} = g^4\left(-E_0^{(4)}\delta_{ll'} - \frac{V_{ln}V_{nn'}V_{nn''}V_{n''l'}}{E_n E_{n'} E_{n''}} + \frac{V_{ln}V_{nl''}}{E_n^2}\frac{V_{l''n''}V_{n''l'}}{E_{n''}}\right),
\tag{C.16}
$$

in exact agreement with (C.9).

## C.2 Cancellation of disconnected pieces at $O(g^4)$

Finally, we would like to explicitly verify at $O(g^4)$ the argument made in (9) that the disconnected pieces of the correlator vanish due to the vacuum energy subtraction. Start with (C.9).

In all cases, the indices $l, l'$ are not summed over and these states must be created by operators $\mathcal{O}_l, \mathcal{O}_{l'}$ in the correlator, as in (16). On the other hand, the states $l'', l''', n, n', n''$ are summed over and arise from the insertion of the identity as a sum over states, rather than from an explicit insertion of the external operator. Now, in general, when we have a string of $n$ $V$s, we can represent it as a correlator of the form

$$\langle \mathcal{O}_l(x) \overbrace{\mathcal{O}_R \ldots \mathcal{O}_R}^{n \text{ times}} \mathcal{O}_{l'} \rangle, \tag{C.17}$$

and if we keep the singular terms in the OPE, we get one disconnected contribution where $\mathcal{O}_l$ and $\mathcal{O}_{l'}$ contract to produce a factor of $\delta_{ll'}$ times some factor $\Lambda$ from doing the Fourier integration, and $n$ factors of $M_{ll'}$ where $M_{ll'}$ is the integral in the text of the leading OPE coefficient $m_{ll'}$ for $\mathcal{O}_l \mathcal{O}_R \mathcal{O}_{l'} \sim \mathcal{O}_R$. There is one more important step, which is that we need to argue that we can move the operators $\mathcal{O}_l$ through the restriction onto light or heavy states. The key point is that this restriction does not break up any irreducible representations of the chiral algebra – all terms in the sum are over complete irreducible representations. Because the basis states are themselves chiral, they therefore commute with the projections onto light and heavy states, e.g.

$$\begin{aligned} \sum_n |n\rangle \langle n| T(z) &= \sum_n \sum_m |n\rangle \langle n| \frac{L_m}{z^{m+2}} = \sum_m z^{-m-2} \sum_n |n\rangle \langle n+m| \\ &= \sum_m z^{-m-2} \sum_n |n-m\rangle \langle n| = \sum_m z^{-m-2} \sum_n L_m |n\rangle \langle n| \\ &= T(z) \sum_n |n\rangle \langle n|, \end{aligned} \tag{C.18}$$

for any sum over $n$ within a full Virasoro representation.

Therefore, we can make the replacement at large $p$

$$\langle \mathcal{O}_l(x) \overbrace{\mathcal{O}_R \ldots \mathcal{O}_R}^{n \text{ times}} \mathcal{O}_{l'} \rangle \to (n M_{ll'} + \delta_{ll'}) \langle \overbrace{\mathcal{O}_R \ldots \mathcal{O}_R}^{n \text{ times}} \rangle. \tag{C.19}$$

After making this replacement, (C.9) simplifies to

$$(P_+)_{\text{eff}}|_{g^4} = g^4 \left( E_0^{(4)} \delta_{ll'} - (4M_{ll'} + \delta_{ll'}) \left( \frac{V_{0n} V_{nn'} V_{n'n''} V_{n''0}}{E_n E_{n'} E_{n''}} \right) + (4M_{ll'} + \delta_{ll'}) \left( \frac{V_{0n} V_{nl''} V_{l''n'} V_{n'0}}{E_n^2 E_{n'}} \right) \right). \tag{C.20}$$

Because of momentum conservation, the only modes that contribute to the sum on $l''$ in the last line are the chiral modes with zero momentum, but the only such mode is the vacuum. Therefore, we can set $l''' = 0$. Finally, we can use the standard formula from time-independent perturbation theory for $E_0^{(4)}$:

$$E_0^{(4)} = -\frac{V_{0n} V_{nn'} V_{n'n''} V_{n''n}}{E_n E_{n'} E_{n''}} + \frac{V_{0n} V_{n0} V_{0n'} V_{n'0}}{E_n^2 E_{n'}}, \tag{C.21}$$

and we see

$$\begin{aligned} (P_+)_{\text{eff}}|_{g^4} &= g^4 (-E_0^{(4)} \delta_{ll'} - (4M_{ll'} + \delta_{ll'})(-E_0^{(4)}) \\ &= g^4 4 M_{ll'} E_0^{(4)} \\ &= M_{ll'} g \frac{d}{dg} (g^4 E_0^{(4)}), \end{aligned} \tag{C.22}$$

and all the disconnected terms (i.e., the terms proportional to $\delta_{ll'}$, where the two external states contract with each other) have cancelled.

# D  IR regulator for the fermion mass term

The fermion mass term has an IR divergence. This is most easily seen in the two particle sector

$$\langle\Psi|m^2\psi\frac{1}{\partial}\psi|\Psi'\rangle = m^2\int_0^1 dx\,\Psi(x,1-x)\Psi'(x,1-x)\left(\frac{1}{x}+\frac{1}{1-x}\right). \tag{D.1}$$

For states created by generic operators, for example, $\mathcal{O}=\psi\partial\psi$, the wave function goes like $\Psi(x,1-x)\sim\mathcal{O}(1)$ at $x\to 0$ or $1$, and the matrix elements have a log divergence. This IR divergence means certain combinations of basis states are infinitely heavy and stay in the UV. Not all states are infinitely heavy, as one can construct a new basis of states where the boundary condition that causes the divergence is eliminated, and the corresponding mass matrix elements are finite. Such basis is known as the *Dirichlet basis* [4]. In the IFT case, we have both the $\sigma$ deformation which does not require a Dirichlet basis, and the $\epsilon$ deformation which does. The situation is similar to 2D QCD with nonzero quark mass [16], where the Coulomb potential term does not require Dirichlet basis and the quark mass term does. In principle, choosing a uniform Dirichlet basis with boundary condition $\Psi(x,\cdots)\sim x$ at $x\to 0$ guarantees IR finiteness, but in practice the corresponding Hamiltonian converges much slower at small mass. In [16], we found that taking a different boundary condition $\Psi(x,\cdots)\sim x^\alpha$ at $x\to 0$ with $\alpha>0$ also satisfies IR finiteness, and fast convergence can be achieved by choosing an optimal $\alpha$ that depends on $m$. We expect that an alternative Dirichlet boundary condition should also be the correct and optimal setup for IFT. However, this approach requires deriving new matrix elements. Instead, we use a numerical trick that also gives us the correct results and fast convergence. The trick is to take an IR regulator that trims off the wave function for $x\le\epsilon$ for each particle, where $\epsilon$ is small but finite, and to take $\epsilon$ to zero as we take $\Delta_{\max}$ to infinity. It turns out that there exists a "Goldilocks" IR regulator $\epsilon$ for each $\eta$ that makes the convergence fast and gives us correct numerical results. In this appendix, we present the details of the Goldilocks IR regulator.

The mass term matrix elements are computed from the Fourier transformation of the following Wick contraction result

$$\langle\partial^{\boldsymbol{k}}\psi|\psi\frac{1}{\partial}\psi|\partial^{\boldsymbol{k}'}\psi\rangle \supset \frac{\Gamma(k+1)}{(x-y)^{k-1}}\frac{1}{i\partial}\frac{\Gamma(k'+1)}{(y-z)^{k'-1}}\frac{A_{\boldsymbol{k}/k,\boldsymbol{k}'/k'}}{(x-z)^c}, \tag{D.2}$$

where $c=\Delta-(k+\frac{1}{2})+\Delta'-(k'+\frac{1}{2})$ captures all spectators. The matrix element is the Fourier transformation of this spacial factor. In order to carry out the Fourier transform, we take the trick of first Fourier transform the individual spacial factors

$$\begin{aligned}
\frac{1}{(x-y)^{k-1}} &= \int dp_1\frac{p_1^k e^{-ip_1(x-y)}}{\Gamma(k+1)}, \\
\frac{1}{(y-z)^{k'-1}} &= \int dp_2\frac{p_2^{k'} e^{-ip_2(y-z)}}{\Gamma(k'+1)}, \\
\frac{1}{(x-z)^c} &= \int dq\frac{q^{c-1} e^{-iq(x-z)}}{\Gamma(c)},
\end{aligned} \tag{D.3}$$

and turn the integral into a momentum space convolution

$$
\begin{aligned}
\int & e^{i(px-p'z)}dxdydz\,\frac{\Gamma(k+1)}{(x-y)^{k-1}}\frac{1}{i\partial}\frac{\Gamma(k'+1)}{(y-z)^{k'-1}}\frac{A_{\boldsymbol{k}/k,\boldsymbol{k}'/k'}}{(x-z)^c} \\
&= \int dp_1 dp_2 dq\, e^{i(px-p'z)}dxdydz\,\frac{1}{p_2}\frac{p_1^k e^{-ip_1(x-y)}}{\Gamma(k+1)}\frac{p_2^{k'} e^{-ip_2(y-z)}}{\Gamma(k'+1)}\frac{q^{c-1}e^{-iq(x-z)}}{\Gamma(c)} \\
&\quad \times\, \Gamma(k+1)\Gamma(k'+1)A_{\boldsymbol{k}/k,\boldsymbol{k}'/k'} \\
&= \int dp_1 dp_2 dq\,(2\pi)^3\delta(p_1-p_2)\delta(p-p_1-q)\delta(p'-p_2-q) \\
&\quad \times\, \frac{p_1^k p_2^{k'} q^{c-1}}{p_2}\frac{A_{\boldsymbol{k}/k,\boldsymbol{k}'/k'}}{\Gamma(c)}\,.
\end{aligned}
\tag{D.4}
$$

The delta functions fixes the parameterization

$$
p_1 = p_2 = px\,,
\tag{D.5}
$$
$$
q = p(1-x)\,,
\tag{D.6}
$$

and the overall momentum dependence $p^{k+k'+c-1}\delta(p-p')$ can be factored out, leaving an integral in the parton momentum fraction $x$. The final integral is

$$
\mathcal{M}_{\boldsymbol{k},\boldsymbol{k}'} \supset \frac{A_{\boldsymbol{k}/k,\boldsymbol{k}'/k'}}{\Gamma(c)}\int dx\,\frac{x^{(k+k')}(1-x)^{c-1}}{x}\,,
\tag{D.7}
$$

for a certain active part. The final matrix element $\mathcal{M}_{\boldsymbol{k},\boldsymbol{k}'}$ sums over all choices of active parts.

- If $k + k' \geq 1$, then the above integral is finite

$$
\int dx\,\frac{x^{(k+k')}(1-x)^{c-1}}{x} = \frac{\Gamma(k+k')\Gamma(c)}{\Gamma(k+k'+c)}\,,
\tag{D.8}
$$

  so we obtain the conventional Wick contraction result

$$
\mathcal{M}_{\boldsymbol{k},\boldsymbol{k}'} \supset A_{\boldsymbol{k}/k,\boldsymbol{k}'/k'}\frac{\Gamma(k+k')}{\Gamma(\Delta+\Delta'-1)}\,.
\tag{D.9}
$$

- If $k = k' = 0$, then the integral in (D.7) is singular. The regularization we impose is to truncate the parton $x$ at a small $\epsilon$,

$$
\int_\epsilon^1 dx\,\frac{x^{(k+k')}(1-x)^{c-1}}{x} = -H_{c-1}-\log(\epsilon)\,,
\tag{D.10}
$$

  where $H_n$ is the Harmonic number. We never worry about the other boundary $x \sim 1$ because $c \geq 1$.

Therefore, the regularized IR divergence is conveniently computed as an extra term in the Hamiltonian which only fires when $k = k' = 0$.

A free massive fermion truncated Hamiltonian converges as $\delta p_+ \sim \Delta_{\max}^{-2}$, and free massive particle has energy $p_+ = m^2/(2p_-)$, indicating that the $\Delta_{\max}$ truncation has a finite momentum resolution in $\delta p_- \sim m^2 \Delta_{\max}^{-2}$. This suggests an ansatz for our momentum IR cutoff

$$
\log \epsilon = -2\log\Delta_{\max} + \xi(m)\,,
\tag{D.11}
$$

and the remaining piece $\xi(m)$ does not depend on $\Delta_{\max}$. We further determine $\xi(m)$ to be the one that results in fastest convergence. In this paper, our best approximation of $\xi(m)$ is found by solving for $\xi(m)$ that gives to $m_1(\Delta_{\max}) = m_1(\Delta_{\max}-2)$ (that is, such that the lowest eigenvalue of the Hamiltonian does not change if we reduce our truncation parameter $\Delta_{\max}$ by $\Delta_{\max} \to \Delta_{\max} - 2$) for our largest truncation $\Delta_{\max} = 36$. Conveniently, $\xi(m)$ enters the Hamiltonian linearly, so that varying $\xi(m)$ is efficient and does not require computing a new set of Hamiltonian matrix elements every time we change its value.

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
