# Peer review of "Lightcone Hamiltonian for Ising Field Theory I: T < T_c"

_SciPost Physics, doi:SciPost Phys. 18, 179 (2025)_

## Round 2 · Referee Report · James Ingoldby (Referee 1) · 2025-2-27

Report

Hamiltonian Truncation is a first-principles, numerical approach for solving interacting quantum theories nonperturbatively. The authors develop a technique in this family called Lightcone Conformal Truncation, which is applicable to QFTs defined as conformal field theories (CFTs) with relevant deformations. In this approach, the lightcone Hamiltonian (generator of translations in lightcone time $t+x$) for the QFT is constructed as an explicit matrix of finite size, that acts on a basis of states built using operators of the undeformed CFT with low scaling dimension from one chiral sector.

The authors seek to extend this method to theories where the operator deforming the CFT can acquire a vacuum expectation value (VEV). The new results of the paper include a compact, analytic expression for the extra term that must be added to the effective lightcone Hamiltonian in this case (given in equation 2.9), and numerical demonstrations of their effective Hamiltonian for the 2d Ising Field Theory. In particular, they compute the low-lying spectrum and matrix elements of the stress energy tensor and compare them with exact results from integrability (along a slice of the model parameter space), and with earlier results from equal-time Hamiltonian Truncation.

The numerical results were obtained using significantly smaller bases of states than were needed in the equal-time HT results shown for comparison. This alone provides a clear demonstration of the advantages of their method. The authors also helpfully provide Mathematica software for building the effective Hamiltonian along with their paper, lowering the barrier to entry for others to apply their framework. However, a key question for future work is whether their approach can be readily generalized. The effective Hamiltonian in equation 2.9 was derived under the assumption of a single relevant deformation with scaling dimension $\Delta<1$. If a similarly compact expression cannot be found without imposing these conditions, the applicability of the method may be significantly restricted.

Overall, this paper meets the journal’s requirement for opening a new pathway in an existing research direction, with clear potential for follow-up work, at least in a particular class of 2d QFTs. I would request only a few minor revisions to clarify the presentation:

  1. The equation 2.9 is derived from equation 1.2 within the paper, but the derivation of the start point, 1.2, is only presented in references. I think that including some more information from the earlier works would clarify the logical flow. Ref [4] suggested that Eq. 1.2 is equivalent to the “Schur’s complement formulation” of the effective Hamiltonian (also given in Eq. C.4 in the current paper) at all orders in perturbation theory - and not just up to O(g^4). Including this information explicitly in the introduction would be helpful, I think. Also, it seemed as if 1.2 was derived originally in 1803.10793, which was not in the reference list (perhaps intentionally). Either way, including more detail on which principles were being used to derive 1.2, as well as more precise guidance as to where to find the clearest, most fundamental derivation of 1.2 from the literature, might be useful.

  2. Following on from the previous point: On page 3, a reference should be given following the statement “As we argued in our previous paper…” in the block of text following equation 1.2.

  3. What power counting rule is being employed in writing down the effective Hamiltonian? Effective Field Theories must be organised as expansions in small parameters, where each parameter is some low energy scale divided by a high energy scale. Could the authors clarify the power counting scheme being used in Eq. 1.2? It would be helpful to specify the small parameters involved and how many orders are retained in the expansion.

  4. What contour is $x$ being integrated over in equation 1.3? Also, is it a lightcone $x^-$ or the periodic spatial coordinate $x$ of the cylinder?

  5. Why can we not study the other half of the phase diagram (where $T>T_c$) with the current approach? Is it due to needing states from the fermionic chiral sector (as well as the bosonic) to describe this phase, or is it because the epsilon operator gets a VEV? A further comment about how to generalise to this case would clarify this.

  6. When comparing data for the spectrum obtained with lightcone conformal truncation and equal time Hamiltonian Truncation, is the idea that the lightcone data is automatically in the infinite volume limit, whereas the equal time data should be extrapolated to the infinite volume limit first before making the comparison? It might be useful to include the volumes used in the data taken from Ref [8] and mention this in the discussion of Figure 5. Also, what units are the masses on the y-axis of Figure 5 measured in?

Recommendation

Ask for minor revision

  • validity: -
  • significance: -
  • originality: -
  • clarity: -
  • formatting: -
  • grammar: -

Author:  Yuan Xin  on 2025-05-02  [id 5437]

(in reply to Report 1 by James Ingoldby on 2025-02-27)
Category:
answer to question

We thank the referee for reviewing our manuscript thoroughly and providing a detailed report. We have made the minor revisions according to referee's suggestions. We answer the referee's questions as follows:

  • The equation 2.9 is derived from equation 1.2 within the paper, but the derivation of the start point, 1.2, is only presented in references. I think that including some more information from the earlier works would clarify the logical flow. Ref [4] suggested that Eq. 1.2 is equivalent to the “Schur’s complement formulation” of the effective Hamiltonian (also given in Eq. C.4 in the current paper) at all orders in perturbation theory - and not just up to O(g^4). Including this information explicitly in the introduction would be helpful, I think. Also, it seemed as if 1.2 was derived originally in 1803.10793, which was not in the reference list (perhaps intentionally). Either way, including more detail on which principles were being used to derive 1.2, as well as more precise guidance as to where to find the clearest, most fundamental derivation of 1.2 from the literature, might be useful.

We have added a footnote saying what the key insight was in ref [2306.13171] compared to [1803.10793], and it is the definition in ref [2306.13171] which is used in this paper. We also mention in the footnote that ref [2306.13171] proved the newer definition is equivalent to what one would get from using the Schur complement method.

  • Following on from the previous point: On page 3, a reference should be given following the statement “As we argued in our previous paper…” in the block of text following equation 1.2.

We have added reference [2306.13171] to this line.

  • What power counting rule is being employed in writing down the effective Hamiltonian? Effective Field Theories must be organised as expansions in small parameters, where each parameter is some low energy scale divided by a high energy scale. Could the authors clarify the power counting scheme being used in Eq. 1.2? It would be helpful to specify the small parameters involved and how many orders are retained in the expansion.

The small parameter is $1/(p R) << 1$ in Eq. (1.2) and the effective Hamiltonian keeps it to $O((pR)^{−1})$. The added explanation of (1.2) in the introduction mentioned above also clarifies this point. What is different in this case compared to most EFTs is that, because momentum p is a parameter that one is free to dial within any theory, we can take the limit $p\rightarrow \infty$ and remove all irrelevant contributions.

  • What contour is x being integrated over in equation 1.3? Also, is it a lightcone x− or the periodic spatial coordinate x of the cylinder?

The contour is in coordinate $x^-$ over the entire real line. We have changed (1.3) to show this explicitly.

  • Why can we not study the other half of the phase diagram (where $T > T_c$) with the current approach? Is it due to needing states from the fermionic chiral sector (as well as the bosonic) to describe this phase, or is it because the epsilon operator gets a VEV? A further comment about how to generalise to this case would clarify this.

We are not completely sure of the answer to this important question, but it is likely that we need to turn on other operators or perhaps the fermion sector for $T > T_c$. A candidate of such an operator is the disorder operator \mu which gets a vev in the $T > T_c$ phase. The new operator will connect the boson and fermionic sector through OPE $\mu\psi$ ~ $\sigma$, so the fermionic sector is likely required in this phase as well. We have added a paragraph in the end of the introduction commenting on this.

  • When comparing data for the spectrum obtained with lightcone conformal truncation and equal time Hamiltonian Truncation, is the idea that the lightcone data is automatically in the infinite volume limit, whereas the equal time data should be extrapolated to the infinite volume limit first before making the comparison? It might be useful to include the volumes used in the data taken from Ref [8] and mention this in the discussion of Figure 5. Also, what units are the masses on the y-axis of Figure 5 measured in?

Indeed, the equal time Hamiltonian truncation is studied at finite volume and the result should be extrapolated to the infinite volume limit, whereas lightcone conformal truncation is formally at infinite volume and no such extrapolation is required. The equal time data were drawn from Ref [8] and Ref [3] and there were no visible difference between the spectra plots in the two references. Both references mentioned finite size extrapolation and claim 5~6 digits of precision. The way Ref [8] and Ref [3] extrapolate to infinite volume is not completely clear to us, but in any case because extrapolations were used their final result does not correspond to a single value of the volume in a simple way that we can quote in the paper. We have added a comment in a footnote on page 23.

The units of Fig 5 are such that $g=2\pi$, we thank the referee for catching this omission.

---

## Round 2 · Referee Report · Anonymous (Referee 2) · 2025-2-28

Report

The paper develops an effective Lightcone Hamiltonian for Ising Field Theory in the low-temperature phase (m<0). The authors verify the approach using integrability results, as well as broader applications including bound-state form factors and stress-tensor spectral density.

The paper is interesting, features multiple self-validation crosschecks, and therefore I recommend its publication. However, the clarity of the exposition could be improved, and I have a number of questions that should be addressed as well as few comments.

  • Towards end of page 2, the authors mention “our prior paper”. A citation is needed there. Similarly in page 3, “As we argued in our previous paper”. Which one? It shouldn’t be left to the author to figure out which paper are referring to.
  • Equation (1.5) contains no explanation at all for the first term.
  • Is time ordering missing in 2.2?
  • The discussions arguments around equation (2.5): is this the reason why the assumption Delta < 1 is needed? What is the effect of subleading terms on the OPE? It is never explained at which step the assumption Delta < 1 is needed.
  • Similarly to the previous point, what is the order of the corrections missing in (2.9)?
  • Sections 2.2, 2.3 and the crosschecks of the appendix are useful.
  • The work set units such that the finite volume R=1. Nevertheless it will add clarity to include the limit of integration in the ‘key’ equations (1.5) and (1.2). In the derivation following equation 2.1, perhaps it can be stated that the authors omit limits of integration.
  • Section 3 is very interesting. The comparison and agreement with integrability results is remarkable! The work though uses input from integrability by inputing the vev of sigma in the LC diagonalization. It is unlcear to me how good will be the comparison if <\sigma> was computed numerically, instead of using integrability input. My question is however partially addressed in the next section, when turning on “m”.
  • The comparison with TFFSA/TCSA is very interesting. It is remarkable that the LC isolated the correct degrees of freedom to reproduce the observables computed here. It is correct that the authors did not include Heff corrections in the TFFSA/TCSA comparison? It would be interesting to compare against TFFSA with similar level of sophistication regarding Heff and <signa>.
  • Would it be interesting to extract phase-shifts? Via the asymptotic bethe ansatz it should be quite straightforward from the authors finite volume data. In the integrability limit it is a nice observable to crosscheck.

Recommendation

Publish (meets expectations and criteria for this Journal)

  • validity: -
  • significance: -
  • originality: -
  • clarity: -
  • formatting: -
  • grammar: -

Author:  Yuan Xin  on 2025-05-02  [id 5436]

(in reply to Report 2 on 2025-02-28)
Category:
answer to question

We thank the referee for reviewing our manuscript thoroughly and providing a detailed report. We have made the minor revisions according to referee's suggestion. We answer the referee's questions as follows:

  • Towards end of page 2, the authors mention “our prior paper”. A citation is needed there. Similarly in page 3, “As we argued in our previous paper”. Which one? It shouldn’t be left to the author to figure out which paper are referring to.

We have added reference [2] to these lines.

  • Equation (1.5) contains no explanation at all for the first term.

We have added an explanation of this term, which is not original to this work.

  • Is time ordering missing in 2.2?

Yes (2.2) is time-ordered. All operators are explicitly put in their time-ordered positions, we have clarified that the times of the external operators are chosen as ordered in the expression.

  • The discussions arguments around equation (2.5): is this the reason why the assumption Delta < 1 is needed? What is the effect of subleading terms on the OPE? It is never explained at which step the assumption Delta < 1 is needed.

The referee is correct and we thank them for emphasizing this important point. Indeed, the OPE contains a term of $O(p^{\Delta - 1})$. In the $\Delta > 1$ case this term becomes dominant and the argument breaks down. We have added a footnote explaining this in more detail.

  • Similarly to the previous point, what is the order of the corrections missing in (2.9)?

In (2.9) we have explicitly taken $p \rightarrow \infty$ and therefore there are no corrections. At finite $p$, for $0 < \Delta < 1$, the finite $p$ corrections would include terms suppressed by powers of $p$ as discussed above.

  • The work set units such that the finite volume R=1. Nevertheless it will add clarity to include the limit of integration in the ‘key’ equations (1.5) and (1.2). In the derivation following equation 2.1, perhaps it can be stated that the authors omit limits of integration.

We have added explicit limits of integration for (2.1) and (1.5), where the integrals were over angles from 0 to $2\pi$, and we have clarified that integrals over $x^-$ are from $-\infty$ to $\infty$ unless specifically stated otherwise.

  • Section 3 is very interesting. The comparison and agreement with integrability results is remarkable! The work though uses input from integrability by inputing the vev of sigma in the LC diagonalization. It is unlcear to me how good will be the comparison if <$\sigma$> was computed numerically, instead of using integrability input. My question is however partially addressed in the next section, when turning on “m”.

The referee is correct that in the subsequent section, $\langle \sigma\rangle$ is computed numerically, and so provides a proof of concept that numerical computations of the vev can at least sometimes be accurate enough to not be the dominant source of numeric error. However, we want to emphasize that even if the vev is not known at all, the effective theory is still useful for computing dimensionless ratios, since the sigma vev simply becomes one dimensionless input to the theory (and in fact, $g \sigma/m^2$ is the only dimensionless input to the theory, so that it has exactly the same number of dimensionless model parameters as the equal-time infinite-volume formulation). We have added a comment on page 4 emphasizing this.

  • The comparison with TFFSA/TCSA is very interesting. It is remarkable that the LC isolated the correct degrees of freedom to reproduce the observables computed here. It is correct that the authors did not include Heff corrections in the TFFSA/TCSA comparison? It would be interesting to compare against TFFSA with similar level of sophistication regarding Heff and <sigma>.

It is true that we did not include Heff corrections in the equal time comparison. So far equal time Effective Hamiltonian has not been applied to Ising Field Theory with $\sigma$ turned on. It would be interesting to compare against TFFSA with leading or next-to-leading order effective Hamiltonian.

  • Would it be interesting to extract phase-shifts? Via the asymptotic bethe ansatz it should be quite straightforward from the authors finite volume data. In the integrability limit it is a nice observable to crosscheck.

We thank the referee for their suggestion, however it is not clear to us how we could use Luscher's method to extract the phase shifts from our spectrum since our finite volume results are in the infinite momentum limit, and as far as we know it is not known (and likely not possible since the spectrum becomes continuous) to extract the phase shifts from the infinite momentum energy eigenvalues.

Anonymous on 2025-05-12  [id 5473]

(in reply to Yuan Xin on 2025-05-02 [id 5436])

The authors have addressed all my questions. I recommend the paper for publication.

---

## Round 3 · Author Response

We thank the referees for reviewing our manuscript thoroughly and providing detailed reports. We have made the minor revisions according to referee's suggestions. We answer the referees' questions as follows:

-- For referee report 1

  1. The equation 2.9 is derived from equation 1.2 within the paper, but the derivation of the start point, 1.2, is only presented in references. I think that including some more information from the earlier works would clarify the logical flow. Ref [4] suggested that Eq. 1.2 is equivalent to the “Schur’s complement formulation” of the effective Hamiltonian (also given in Eq. C.4 in the current paper) at all orders in perturbation theory - and not just up to O(g^4). Including this information explicitly in the introduction would be helpful, I think. Also, it seemed as if 1.2 was derived originally in 1803.10793, which was not in the reference list (perhaps intentionally). Either way, including more detail on which principles were being used to derive 1.2, as well as more precise guidance as to where to find the clearest, most fundamental derivation of 1.2 from the literature, might be useful.

We have added a footnote saying what the key insight was in ref [2306.13171] compared to [1803.10793], and it is the definition in ref [2306.13171] which is used in this paper. We also mention in the footnote that ref [2306.13171] proved the newer definition is equivalent to what one would get from using the Schur complement method.

  1. Following on from the previous point: On page 3, a reference should be given following the statement “As we argued in our previous paper…” in the block of text following equation 1.2.

We have added reference [2306.13171] to this line.

  1. What power counting rule is being employed in writing down the effective Hamiltonian? Effective Field Theories must be organised as expansions in small parameters, where each parameter is some low energy scale divided by a high energy scale. Could the authors clarify the power counting scheme being used in Eq. 1.2? It would be helpful to specify the small parameters involved and how many orders are retained in the expansion.

The small parameter is 1/(p R) << 1 in Eq. (1.2) and the effective Hamiltonian keeps it to O((pR)^−1). The added explanation of (1.2) in the introduction mentioned above also clarifies this point. What is different in this case compared to most EFTs is that, because momentum p is a parameter that one is free to dial within any theory, we can take the limit $p\rightarrow \infty$ and remove all irrelevant contributions.

  1. What contour is x being integrated over in equation 1.3? Also, is it a lightcone $x^−$ or the periodic spatial coordinate $x$ of the cylinder?

The contour is in coordinate $x^-$ over the entire real line. We have changed (1.3) to show this explicitly.

  1. Why can we not study the other half of the phase diagram (where $T > Tc$) with the current approach? Is it due to needing states from the fermionic chiral sector (as well as the bosonic) to describe this phase, or is it because the epsilon operator gets a VEV? A further comment about how to generalise to this case would clarify this.

We are not completely sure of the answer to this important question, but it is likely that we need to turn on other operators or perhaps the fermion sector for $T > T_c$. A candidate of such an operator is the disorder operator $\mu$ which gets a vev in the $T > T_c$ phase. The new operator will connect the boson and fermionic sector through OPE $\mu\psi \sim \sigma$, so the fermionic sector is likely required in this phase as well. We have added a paragraph in the end of the introduction commenting on this.

  1. When comparing data for the spectrum obtained with lightcone conformal truncation and equal time Hamiltonian Truncation, is the idea that the lightcone data is automatically in the infinite volume limit, whereas the equal time data should be extrapolated to the infinite volume limit first before making the comparison? It might be useful to include the volumes used in the data taken from Ref [8] and mention this in the discussion of Figure 5. Also, what units are the masses on the y-axis of Figure 5 measured in?

Indeed, the equal time Hamiltonian truncation is studied at finite volume and the result should be extrapolated to the infinite volume limit, whereas lightcone conformal truncation is formally at infinite volume and no such extrapolation is required. The equal time data were drawn from Ref [8] and Ref [3] and there were no visible difference between the spectra plots in the two references. Both references mentioned finite size extrapolation and claim 5~6 digits of precision. The way Ref [8] and Ref [3] extrapolate to infinite volume is not completely clear to us, but in any case because extrapolations were used their final result does not correspond to a single value of the volume in a simple way that we can quote in the paper. We have added a comment in a footnote on page 23.

The units of Fig 5 are such that $g=2\pi$, we thank the referee for catching this omission.

-- For referee report 2

  1. Towards end of page 2, the authors mention “our prior paper”. A citation is needed there. Similarly in page 3, “As we argued in our previous paper”. Which one? It shouldn’t be left to the author to figure out which paper are referring to.

We have added reference [2] to these lines.

  1. Equation (1.5) contains no explanation at all for the first term.

We have added an explanation of this term, which is not original to this work.

  1. Is time ordering missing in 2.2?

Yes (2.2) is time-ordered. All operators are explicitly put in their time-ordered positions, we have clarified that the times of the external operators are chosen as ordered in the expression.

  1. The discussions arguments around equation (2.5): is this the reason why the assumption Delta < 1 is needed? What is the effect of subleading terms on the OPE? It is never explained at which step the assumption Delta < 1 is needed.

The referee is correct and we thank them for emphasizing this important point. Indeed, the OPE contains a term of $O(p^{\Delta - 1})$. In the $\Delta > 1$ case this term becomes dominant and the argument breaks down. We have added a footnote explaining this in more detail.

  1. Similarly to the previous point, what is the order of the corrections missing in (2.9)?

In (2.9) we have explicitly taken $p \rightarrow \infty$ and therefore there are no corrections. At finite $p$, for $0 < \Delta < 1$, the finite $p$ corrections would include terms suppressed by powers of $p$ as discussed above.

  1. The work set units such that the finite volume R=1. Nevertheless it will add clarity to include the limit of integration in the ‘key’ equations (1.5) and (1.2). In the derivation following equation 2.1, perhaps it can be stated that the authors omit limits of integration.

We have added explicit limits of integration for (2.1) and (1.5), where the integrals were over angles from 0 to $2\pi$, and we have clarified that integrals over $x^-$ are from $-\infty$ to $\infty$ unless specifically stated otherwise.

  1. Section 3 is very interesting. The comparison and agreement with integrability results is remarkable! The work though uses input from integrability by inputing the vev of sigma in the LC diagonalization. It is unlcear to me how good will be the comparison if <$\sigma$> was computed numerically, instead of using integrability input. My question is however partially addressed in the next section, when turning on “m”.

The referee is correct that in the subsequent section, $\langle \sigma\rangle$ is computed numerically, and so provides a proof of concept that numerical computations of the vev can at least sometimes be accurate enough to not be the dominant source of numeric error. However, we want to emphasize that even if the vev is not known at all, the effective theory is still useful for computing dimensionless ratios, since the sigma vev simply becomes one dimensionless input to the theory (and in fact, $g \sigma/m^2$ is the only dimensionless input to the theory, so that it has exactly the same number of dimensionless model parameters as the equal-time infinite-volume formulation). We have added a comment on page 4 emphasizing this.

  1. The comparison with TFFSA/TCSA is very interesting. It is remarkable that the LC isolated the correct degrees of freedom to reproduce the observables computed here. It is correct that the authors did not include Heff corrections in the TFFSA/TCSA comparison? It would be interesting to compare against TFFSA with similar level of sophistication regarding Heff and <sigma>.

It is true that we did not include Heff corrections in the equal time comparison. So far equal time Effective Hamiltonian has not been applied to Ising Field Theory with $\sigma$ turned on. It would be interesting to compare against TFFSA with leading or next-to-leading order effective Hamiltonian.

  1. Would it be interesting to extract phase-shifts? Via the asymptotic bethe ansatz it should be quite straightforward from the authors finite volume data. In the integrability limit it is a nice observable to crosscheck.

We thank the referee for their suggestion, however it is not clear to us how we could use Luscher's method to extract the phase shifts from our spectrum since our finite volume results are in the infinite momentum limit, and as far as we know it is not known (and likely not possible since the spectrum becomes continuous) to extract the phase shifts from the infinite momentum energy eigenvalues.

---

## Round 3 · List of Changes

List of Changes

  1. Added a footnote explaining the key insight in ref [2306.13171] compared to [1803.10793].

  2. Added reference [2306.13171] to "our prior paper" and "As we argued in our previous paper" on pages 2 and 3.

  3. Added explanation that the small parameter is $1/(pR) \ll 1$ in Eq. (1.2) and clarified how taking $p\rightarrow\infty$ removes irrelevant contributions.

  4. Added a paragraph at the end of the introduction discussing possible approaches to the $T > T_c$ phase.

  5. Added footnote on page 23 explaining the equal time data extrapolation methodology. Added units ($g=2\pi$) to Fig. 5.

  6. Added explanation for the first term in Eq. (1.5).

  7. Clarified that Eq. (2.2) is time-ordered and that the times of the external operators are chosen as ordered in the expression.

  8. Added footnote 4 explaining why $\Delta < 1$ is needed .

  9. Added a statement of integration limits for Eqs. (2.1) and (1.5), noting angle integrations are from 0 to $2\pi$ and $x^-$ integrals are from $-\infty$ to $\infty$ unless otherwise specified.

  10. Added statement on page 4 emphasizing that even if the vev is not known at all, the effective theory is still useful for computing dimensionless ratios

---

## Editorial Decision

published